# Is Importance Weighting Incompatible with Interpolating Classifiers?

**Ke Alexander Wang**[*]**, Niladri S. Chatterji**[*]**, Saminul Haque, Tatsunori Hashimoto**
Department of Computer Science
Stanford University
{alxwang,niladri}@cs.stanford.edu,{saminulh,thashim}@stanford.edu

## ABSTRACT

Importance weighting is a classic technique to handle distribution shifts. However, prior work has presented strong empirical and theoretical evidence demonstrating that importance weights can have little to no effect on overparameterized neural networks. *Is importance weighting truly incompatible with the training of overparameterized neural networks?* Our paper answers this in the negative. We show that importance weighting fails not because of the overparameterization, but instead, as a result of using exponentially-tailed losses like the logistic or cross-entropy loss. As a remedy, we show that polynomially-tailed losses restore the effects of importance reweighting in correcting distribution shift in overparameterized models. We characterize the behavior of gradient descent on importance weighted polynomially-tailed losses with overparameterized linear models, and theoretically demonstrate the advantage of using polynomially-tailed losses in a label shift setting. Surprisingly, our theory shows that using weights that are obtained by exponentiating the classical unbiased importance weights can improve performance. Finally, we demonstrate the practical value of our analysis with neural network experiments on a subpopulation shift and a label shift dataset. When reweighted, our loss function can outperform reweighted cross-entropy by as much as 9% in test accuracy. Our loss function also gives test accuracies comparable to, or even exceeding, well-tuned state-of-the-art methods for correcting distribution shifts.

## 1 INTRODUCTION

Machine learning models are often evaluated on test data which differs from the data that they were trained on. A classic statistical technique to combat such distribution shift is to *importance weight* the loss function during training (Shimodaira, 2000). This procedure upweights points in the training data that are more likely to appear in the test data and downweights ones that are less likely. The reweighted training loss is an unbiased estimator of the test loss and can be minimized by standard algorithms, resulting in a simple and general procedure to address distribution shift.

Surprisingly, recent papers (Byrd & Lipton, 2019; Xu et al., 2020) have found that importance weighting is ineffective in the current deep learning paradigm, where overparameterized models interpolate the training data or have vanishingly small train loss. In particular, Byrd & Lipton (2019) empirically showed that when no regularization is used, overparameterized linear and nonlinear models trained with the importance weighted cross-entropy loss ignore the importance weights. Xu et al. (2020) followed up and provided a theoretical justification for this observation in overparameterized linear and non-linear models.

To build intuition about why importance weighting fails, consider linear classifiers as an example. Given linearly separable data $(x_1, y_1), \ldots, (x_n, y_n) \in \mathbb{R}^d \times \{-1, 1\}$, Soudry et al. (2018) showed that if gradient descent is applied to minimize an exponentially-tailed classification loss $\left(\sum_{i \in [n]} \ell_{\exp}(y_i x_i)\right)$ then the iterates converge in direction to the maximum margin classifier $\widehat{\theta}_{\mathsf{MM}} := \arg\min_{\|\theta\|=1} \{\gamma \, : \, y_i x_i \cdot \theta \geq \gamma, \text{ for all } i \in [n]\}$. Xu et al. (2020) showed that in this same setting, minimizing the importance weighted loss $\left(\sum_{i \in [n]} w_i \ell_{\exp}(y_i x_i)\right)$ with gradient descent also results

---

[*]Equal contribution.

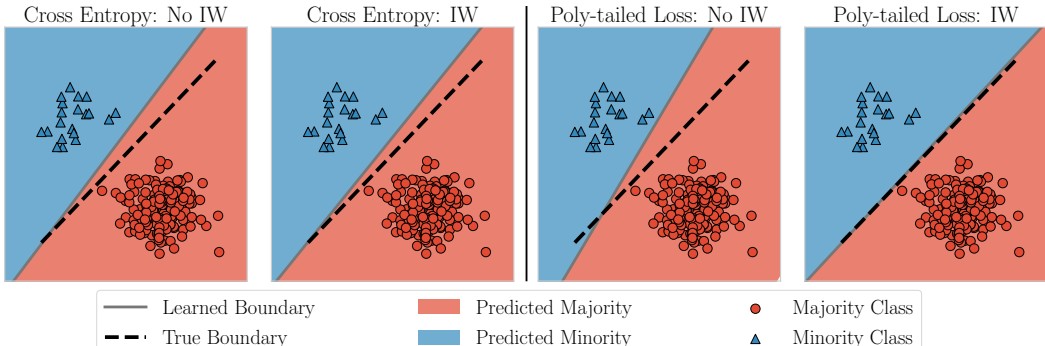

Figure 1: Models trained with gradient descent, with and without importance weights (IW), in the label shift setting where classes are imbalanced in the training set. All models interpolate the training points with 100% accuracy. (Left) Importance weights provably fails to correct for the distribution shift for the cross-entropy loss. The learned boundary is asymptotically the maximum-margin classifier even with reweighting. (Right) Our polynomially-tailed loss restores the effects of importance weights, correctly adjusting for the distribution shift.

in convergence to the maximum margin classifier, regardless of the weights. To see why, consider the special case where the weights $(w_1, \ldots, w_n)$ are positive integers. This reweighting is equivalent to simply repeating each datapoint $w_i$ times, and the maximum margin classifier over this "new dataset" remains unchanged. Thus, invoking the original result by Soudry et al. (2018) proves that the importance weights has no effect in correcting the distribution shift. This result can be seen in Figure 1 where we demonstrate this phenomenon in a simple toy problem.

Such evidence has led some to wonder if importance weighting is fundamentally incompatible with overparameterized interpolating models. In this paper, we show that this is not the case. We find that the culprit behind the ineffectiveness of importance weighting is the *exponential tail* of popular losses such as the cross-entropy or the logistic. We propose altering the structure of the loss to have fatter, polynomially decaying tails instead. We theoretically and empirically show that importance weights do correct for distribution shift under such losses even for overparameterized classifiers.

Our first contribution is to characterize the limiting direction of the iterates of gradient descent (its implicit bias) when minimizing reweighted polynomially-tailed losses with linear classifiers. We show that this limiting direction is a function of both the datapoints as well as the importance weights, unlike the maximum margin classifier that only depends on the data (see the right half of Figure 1). Next, we analyze the generalization behavior of this classifier in a label shift setting. We prove that when the weights are an exponentiation of the unbiased importance weights, the test error decays to zero in the large sample limit, regardless of the level of imbalance in the data. In contrast, we prove that the maximum margin classifier test error in this same setting must be *at least* $1/8$.

Finally, we demonstrate the practical benefits of our framework by applying this approach to experiments with neural networks. In both a label shift dataset (imbalanced binary CIFAR10), and a subpopulation shift dataset with spurious correlations (CelebA (Sagawa et al., 2019)), we find that reweighting polynomially-tailed losses consistently outperforms reweighted cross-entropy loss, as our linear theory suggests. Additionally, poly-tailed loss with biased importance weights can perform comparably to, or better than, state-of-the-art methods distribution shift (Cao et al., 2019; Ye et al., 2020; Menon et al., 2020; Kini et al., 2021)[1].

## 2 RELATED WORK

Early work (Shimodaira, 2000; Wen et al., 2014) already warned against the potential ineffectiveness of importance weights on interpolating overparameterized models. Shimodaira (2000) showed that when the model is well-specified, importance weights can fail to have an effect, and that the ordinary maximum likelihood estimate is asymptotically optimal. Wen et al. (2014) showed that when there is a zero-loss minimizer of an unweighted convex loss minimization problem, then it is also a minimizer of the (adversarially) reweighted loss as well. Recent work (Byrd & Lipton, 2019; Xu et al., 2020) has shown that importance weighting fails to have an effect on neural networks

---

[1]Code is available at https://github.com/KeAWang/importance-weighting-interpolating-classifiers

trained with gradient descent, though always in the setting of exponentially-tailed losses. Sagawa et al. (2019) demonstrated that reweighting can fail to have the desired effect when unregularized distributionally robust optimization (DRO) methods are used in conjunction with the cross-entropy loss. They empirically showed that regularization is necessary to reap the benefits of reweighting, also observed by Byrd & Lipton (2019).

A recent line of work (Cao et al., 2019; Ye et al., 2020; Menon et al., 2020; Kini et al., 2021) has introduced modifications to the logistic and cross-entropy losses to correct for distribution shift. Cao et al. (2019) and Menon et al. (2020) proposed using additive corrections to the logits. However, without regularization or early-stopping these corrections are ineffective since the additive corrections to the logits is analogous to importance weighting exponential-tailed losses. Multiplicative logit corrections (Ye et al., 2020), possibly combined with additive corrections (Kini et al., 2021), have also been proposed. Unlike the additive corrections, these methods do not converge to the max-margin classifier, but they also do not correspond to importance weighting algorithms. In our work, we focus on the question of whether importance weighting alone can correct for distribution shift, and show practical empirical benefits relative to multiplicative logit corrections.

Our work also connects to literature that has studied the implicit bias of gradient descent (Soudry et al., 2018; Ji & Telgarsky, 2019; Nacson et al., 2019). Especially relevant is the work by Ji et al. (2020) who relate the implicit bias of gradient descent with exponentially and polynomially-tailed losses for linear classifiers to a solution of a regularized loss minimization problem. Finally, our generalization analysis draws from the growing literature focused on finite sample bounds on the test error of the maximum margin classifier in the overparameterized regime (Chatterji & Long, 2021; Muthukumar et al., 2021; Wang & Thrampoulidis, 2021; Cao et al., 2021).

## 3 SETTING

We consider a distribution shift setting where the training samples $\{(x_1, y_1), \ldots, (x_n, y_n)\} \in \mathbb{R}^d \times \{-1, 1\}$ are drawn i.i.d. from $\mathsf{P}_{\text{train}}$, and the test samples are drawn from a different distribution $\mathsf{P}_{\text{test}}$ that is absolutely continuous with respect to $\mathsf{P}_{\text{train}}$. Let $f_\theta$ denote a classifier parameterized by $\theta$. Given a feature $x$, a classifier maps this feature to $f_\theta(x) \in \mathbb{R}$. In this paper we shall consider cases where the classifier is either linear (for our theory) or a neural network (for our experiments).

Our goal is to find a classifier $f_\theta$ that minimizes the 0-1 loss with respect to the test distribution:
$$\mathsf{TestError}[f_\theta] = \mathbb{P}_{(x,y) \sim \mathsf{P}_{\text{test}}} \left[ \text{sign}(f_\theta(x)) \neq y \right].$$

To handle the mismatch between $\mathsf{P}_{\text{train}}$ and $\mathsf{P}_{\text{test}}$, we shall study importance weighting algorithms. Given a datapoint $(x, y) \in \mathbb{R}^d \times \{-1, 1\}$, the classical *unbiased importance weight* at $(x, y)$ is given by the ratio of densities between the test and the train distributions $\frac{\mathsf{P}_{\text{test}}(x,y)}{\mathsf{P}_{\text{train}}(x,y)}$. Using these unbiased importance weights ensures that the reweighted training loss is an unbiased estimate of the test loss.

However, as noted above, past work has shown that interpolating classifiers trained with gradient descent on importance weighted exponentially-tailed losses (such as the logistic loss $\ell_{\log}(z) := \log(1 + \exp(-z))$, the exponential loss $\ell_{\exp}(z) := \exp(-z)$, and the cross-entropy loss) ignore the importance weights. For example, consider the case when the classifier is linear $f_\theta(x) = x \cdot \theta$, the weights are $w_1, \ldots, w_n > 0$, and the reweighted loss function is $\widehat{L}(\theta) = \sum_{i=1}^n w_i \ell_{\log}(y_i x_i \cdot \theta)$. Xu et al. (2020) showed that if the data is linearly separable then the iterates of gradient descent converge in direction to the $\ell_2$-maximum margin classifier,
$$\widehat{\theta}_{\mathsf{MM}} := \underset{\theta: \|\theta_2\| = 1}{\arg\max} \left\{ \gamma \; : \; y_i x_i \cdot \theta \geq \gamma \text{ for all } i \in [n] \right\}. \tag{1}$$

Observe that the maximum margin classifier does not depend on the importance weights $(w_1, \ldots, w_n)$ and hence may suffer large test error when there is distribution shift. Xu et al. (2020) further showed that when separability assumptions hold, non-linear classifiers trained with gradient descent on exponentially-tailed losses are also unaffected by importance weights.

We initiate a study of polynomially-tailed losses in the distribution shift setting and show that they have improved behavior with respect to importance weighting even when the model is overparameterized. Given parameters $\alpha > 0$ and $\beta \in \mathbb{R}$ define the polynomially-tailed loss as follows:
$$\ell_{\alpha,\beta}(z) := \begin{cases} \ell_{\text{left}}(z) & \text{if } z < \beta \\ \frac{1}{[z-(\beta-1)]^\alpha} & \text{if } z \geq \beta, \end{cases}$$

where $\ell_{\text{left}}$ is any loss function such that the overall loss function $\ell_{\alpha,\beta}$ is convex, differentiable and strictly decreasing. Several natural choices for $\ell_{\text{left}}$ include the scaled logistic ($c_1 \log(1 + \exp(-c_2 z))$), exponential ($c_1 \exp(-c_2 z)$) or linear ($-c_1 z + c_2$) losses.

Given a training dataset $\{(x_1, y_1), \ldots, (x_n, y_n)\}$ and a set of weights $w_1, \ldots, w_n \geq 0$ we let

$$\widehat{L}_{\alpha,\beta}(f_\theta) := \sum_{i=1}^{n} w_i \ell_{\alpha,\beta}(y_i f_\theta(x_i))$$

be the reweighted empirical loss on this dataset.

**Notation.** Given a vector $v$, let $\|v\|$ denote its Euclidean norm. For any $j \in \mathbb{N}$, we denote the set $\{1, \ldots, j\}$ by $[j]$. A random variable $\xi$ is 1-sub-Gaussian if for any $\lambda \in \mathbb{R}$, $\mathbb{E}\left[e^{\lambda \xi}\right] \leq e^{\lambda^2/2}$.

## 4 THEORETICAL RESULTS

In this section, we present several theoretical results that justify the use of polynomially-tailed losses in conjunction with importance weights to handle distribution shifts. Our final result shows a strict separation between the performance of reweighted polynomially- and exponentially-tailed models under label shift. We restrict our theoretical analysis to linear classifiers, $f_\theta(x) = x \cdot \theta$, for some $\theta \in \mathbb{R}^d$.

First, in Section 4.1 we shall characterize the limiting direction of gradient descent on reweighted polynomially-tailed losses and show that this direction depends on both the weights as well as the datapoints. Next, in Section 4.2, we upper bound the test error of this limiting solution in a label shift setting. We also show that choosing weights that are obtained by exponentiating the unbiased importance weights helps in reducing the test error. Finally, in this label shift setting, we show that the maximum margin classifier suffers an error that is at least $1/8$.

### 4.1 IMPLICIT BIAS OF GRADIENT DESCENT ON POLYNOMIALLY-TAILED LOSSES

We begin by presenting a result that characterizes the implicit bias of gradient descent on reweighted polynomially-tailed losses for linearly separable classification. Understanding this situation is a key first step, as gradient descent for separable linear classification is often used as a simplified model to theoretically characterize the behavior of overparameterized neural networks (Soudry et al., 2018; Ji & Telgarsky, 2019; Nacson et al., 2019; Lyu & Li, 2019; Ji & Telgarsky, 2020).

Given a linearly separable dataset $(x_1, y_1), \ldots, (x_n, y_n)$, we let $z_i := y_i x_i$. We shall analyze the iterates of gradient descent with step-size $\eta > 0$ and initial iterate $\theta^{(0)} \in \mathbb{R}^d$, for all $t \in \{0, 1, \ldots\}$: $\theta^{(t+1)} = \theta^{(t)} - \eta \nabla \widehat{L}_{\alpha,\beta}(\theta^{(t)})$. Define the direction

$$\widehat{\theta}_\alpha := \underset{\theta:\|\theta\|=1}{\arg\min} \left\{ \sum_{i=1}^{n} \frac{w_i}{(z_i \cdot \theta)^\alpha}, \quad \text{s.t.} \quad z_i \cdot \theta > 0, \quad \text{for all } i \in [n] \right\}. \tag{2}$$

The following proposition characterizes the limiting direction of gradient descent iterates.

**Proposition 4.1.** *Suppose that the data is linearly separable. For any $\alpha > 0$, $\beta \in \mathbb{R}$, any initial point $\theta^{(0)} \in \mathbb{R}^d$, and for all small enough step-sizes $\eta$ the direction of the gradient descent iterates satisfy the following:* $\lim_{t \to \infty} \frac{\theta^{(t)}}{\|\theta^{(t)}\|} \to \widehat{\theta}_\alpha$.

The proof is presented in Appendix A. This proof relies recent result by Ji et al. (2020) that relates the limiting direction of gradient descent on the unregularized loss to the limiting solution of a norm-constrained loss minimization problem, where the limit is taken with respect to the norm constraint.

Note that, unlike the maximum margin classifier, it is immediately clear that this limiting direction $\widehat{\theta}_\alpha$ depends on the weights $w_1, \ldots, w_n$. As one would intuitively expect, the direction $\widehat{\theta}_\alpha$ tries to achieve a larger margin $z_i \cdot \theta$ on points with larger weights $w_i$. This behavior is also apparent in the simulation in the rightmost panel in Figure 1, where upweighting points in the minority class helps to learn a classifier that is similar in direction to the Bayes optimal classifier for the problem.

Our limiting direction $\widehat{\theta}_\alpha$ has the interesting property that it does not depend on several quantities: the initial point $\theta^{(0)}$, the properties of $\ell_{\text{left}}$, and the "switchover" point $\beta$. Linear separability ensures

that in the limit, the margin on each point is much larger than $\beta$, and so the loss of each point is in the polynomial tail of part of $\ell_{\alpha,\beta}$.

## 4.2 GENERALIZATION ANALYSIS

The result in the previous subsection shows that the asymptotic classifier learnt by gradient descent, $\widehat{\theta}_\alpha$ respects importance weights. However, this does not demonstrate that using polynomially-tailed losses leads to classifiers with lower test error compared to using exponentially-tailed losses.

To answer this more fine-grained question, we will need to perform a more refined analysis. To do this we will make sub-Gaussian cluster assumptions on the data, similar to ones considered in the generalization analysis of the overparameterized maximum margin classifiers (Chatterji & Long, 2021; Wang & Thrampoulidis, 2021; Liang & Recht, 2021; Cao et al., 2021). In our setting, the features associated with each label are drawn from two different sub-Gaussian clusters. We shall consider a label shift problem where the training data is such that the number of data points from the positive (majority) cluster will be much larger than the number of datapoints from the negative (minority) cluster. The test datapoints will be drawn uniformly from either cluster.

Under these assumptions, we derive upper bounds on the error incurred by $\widehat{\theta}_1$, the limiting direction of gradient descent of polynomially tailed losses with $\alpha = 1$ (Section 4.2.2). We will find that its test error is small when the reweighting weights are set to cubic powers of the unbiased importance weights. (A similar analysis can also be conducted for other $\alpha$.) In the same setting in Section 4.2.3, we will show that the maximum margin classifier must suffer large test errors.

### 4.2.1 SETTING FOR GENERALIZATION ANALYSIS

Here we formally describe the setting of our generalization analysis. We let $C \geq 1$ and $0 < \widetilde{C} \leq 1$ denote positive absolute constants, whose value is fixed throughout the remainder of the paper. We will use $c, c', c_1, \ldots$ to denote "local" positive constants, which may take different values in different contexts.

The training dataset $\mathcal{S} := \{(x_1, y_1), \ldots, (x_n, y_n)\}$ has $n$ independently drawn samples. The conditional distribution of the features given the label is

$$x \mid \{y = 1\} = \mu_1 + Uq$$
$$x \mid \{y = -1\} = \mu_2 + Uq,$$

where $\mu_1 \cdot \mu_2 = 0$, $\|\mu_1\| = \|\mu_2\|$, $U$ is an arbitrary orthogonal matrix, and $q$ is a random variable such that: its entries are 1-sub-Gaussian and independent, and $\mathbb{E}\left[\|q\|^2\right] \geq \widetilde{C}d$.

We note that in past work that studied this setting (Chatterji & Long, 2021; Wang & Thrampoulidis, 2021; Cao et al., 2021), the cluster centers were chosen to be opposite one another $\mu_1 = -\mu_2$. Here, since our goal here is to study a label shift setting we consider the cluster centers $\mu_1$ and $\mu_2$ to be orthogonal. This ensures that learning the direction of one of the centers reveals no information about the other center, which makes this problem more challenging in this setting.

Define $\mathcal{P} := \{i \in [n] : y_i = 1\}$ to be the set of indices corresponding to the positive labels and $\mathcal{N} := \{i \in [n] : y_i = -1\}$ to be the set of indices corresponding to the negative labels. As stated above, we will focus on the case where $|\mathcal{P}| \gg |\mathcal{N}| \geq 1$. Let $\tau := \frac{|\mathcal{P}|}{|\mathcal{N}|} \geq 1$ be the ratio between the number of positive and negative samples. The test distribution $\mathsf{P}_{\mathsf{test}}$ is balanced. That is, if $(x, y) \sim \mathsf{P}_{\mathsf{test}}$, then $\mathbb{P}[y = 1] = \mathbb{P}[y = -1] = 1/2$ and $x \mid y$ follows the distribution as described above. We shall study the case where negative examples (which are in the minority) are upweighted. Specifically, set the importance weights as follows: $w_i = 1$ for $i \in \mathcal{P}$ and $w_i = w$ for $i \in \mathcal{N}$.

**Assumptions.** Given a failure probability $0 < \delta < 1/C$, we make the following assumptions on the parameters of the problem:

1. Number of samples $n \geq C \log(1/\delta)$.
2. Norm of the means $\|\mu\|^2 := \|\mu_1\|^2 = \|\mu_2\|^2 \geq Cn^2 \log(n/\delta)$.
3. Dimension $d \geq Cn\|\mu\|^2$.

Our assumptions allow for large overparameterization, where the dimension $d$ scales polynomially with the number of samples $n$.

### 4.2.2 UPPER BOUND FOR THE REWEIGHTED POLYNOMIALLY-TAILED LOSS CLASSIFIER

First, we shall prove an upper bound on the test error of the solution learnt by gradient descent on the reweighted polynomially-tailed loss. Under the choice of weights described above, by equation (2)

$$\widehat{\theta}_1 = \arg\min_{\theta:\|\theta\|=1} \sum_{i\in\mathcal{P}} \frac{1}{z_i \cdot \theta} + \sum_{i\in\mathcal{N}} \frac{w}{z_i \cdot \theta}, \quad \text{s.t.} \quad z_i \cdot \theta > 0, \quad \text{for all } i \in [n],$$

where $z_i = y_i x_i$ as defined previously. The following theorem provides an upper bound on the test error of $\widehat{\theta}_1$ and specifies a range of values for the weight $w$.

**Theorem 4.2.** *For any $0 < \widetilde{C} \leq 1$, there is a constant $c$ such that, for all large enough $C$ and for any $0 < \delta < 1/C$, under the assumptions of this subsection the following holds. If the weight $\frac{\tau^3}{2} \leq w \leq 2\tau^3$, then with probability at least $1-\delta$, training on $\mathcal{S}$ produces a classifier $\widehat{\theta}_1$ satisfying:*

$$\mathsf{TestError}[\widehat{\theta}_1] = \mathbb{P}_{(x,y)\sim\mathsf{P}_{\text{test}}} \left[ \mathrm{sign}\left(\widehat{\theta}_1 \cdot x\right) \neq y \right] \leq \exp\left( -\frac{cn}{\tau} \frac{\|\mu\|^4}{d} \right).$$

This theorem is proved in Appendix B. To prove this theorem, we first show that the data is linearly separable under our assumptions and use the implicit bias results of Proposition 4.1. Then, our upper bound is equivalent to bounding the test error of the limiting iterate of gradient descent on our reweighted loss. Next, we show that the sum of the gradients across each of the two groups remains roughly balanced throughout training under our choice of $w$. This ensures that the classifier aligns well with both cluster centers $\mu_1$ and $-\mu_2$ which then proves our bound on the test error via a standard Hoeffding bound.

If we consider a regime where $d$ and $n$ are growing simultaneously, then the test error goes down to zero if the norm of the cluster means $\|\mu\|$ grows faster than $d^{1/4}$, regardless of the level of imbalance in the training data as $\tau < n$. Briefly note that in setting with balanced data ($\tau = 1$), the bound on the test error here matches the bound obtained for the maximum margin classifier in (Chatterji & Long, 2021; Wang & Thrampoulidis, 2021; Cao et al., 2021) (although these bounds are not directly comparable as the setting here is slightly modified for a label shift). The impossibility result by Jin (2009) shows that these previous results guarantee that learning occurs right up until the information theoretic frontier ($\|\mu\|^2 = \omega(\sqrt{d})$).

Finally, it is interesting to note that the theorem requires the weight $w$ to scale with $\tau^3$ instead of $\tau$ (the unbiased importance weight), which would ensure that the reweighted training loss is an unbiased estimate of the test loss. In our proof, we find that in order to suffer low test error it is important to guarantee that the norm of the gradients across the two groups is roughly balanced throughout training. We show that at any training iteration, the ratio of the sum of the derivative of the weighted losses in the majority and minority clusters scales as $\frac{\tau}{w^{1/3}}$. Thus choosing $w$ to scale with $\tau^3$ ensures that the gradients are balanced across both groups. We verify this in our simulations (Figure 2), and find that $w = \tau^3$ ensures equal test error across both classes, reducing overall error.

### 4.2.3 LOWER BOUND FOR THE MAXIMUM MARGIN CLASSIFIER

In the previous section we derived an upper bound to demonstrate that classifiers trained with polynomially-tailed losses achieve low test error. We will now show that classifiers trained with exponentially-tailed losses have their error lower bounded by $1/8$ in the same setting.

**Theorem 4.3.** *Let $q \sim \mathsf{N}(0, I_{d\times d})$. There exist constants $c$ and $c'$ such that, for all large enough $C$ and for any $0 < \delta < 1/C$, under the assumptions of this subsection the following holds. With probability at least $1-\delta$, training on $\mathcal{S}$ produces a maximum margin classifier $\widehat{\theta}_{\mathsf{MM}}$ satisfying:*

$$\mathsf{TestError}[\widehat{\theta}_{\mathsf{MM}}] = \mathbb{P}_{(x,y)\sim\mathsf{P}_{\text{test}}} \left[ \mathrm{sign}\left(\widehat{\theta}_{\mathsf{MM}} \cdot x\right) \neq y \right] \geq \frac{1}{2} \cdot \Phi\left( -\frac{c\sqrt{n}}{\tau} \cdot \frac{\|\mu\|^2}{\sqrt{d}} \right),$$

*where $\Phi$ is the Gaussian cdf. Furthermore, if the imbalance ratio $\tau \geq c' \frac{\sqrt{n}\|\mu\|^2}{\sqrt{d}}$ then with probability at least $1-\delta$, $\mathsf{TestError}[\widehat{\theta}_{\mathsf{MM}}] \geq \frac{1}{8}$.*

This theorem is proved in Appendix C. Intuitively, the lower bound holds because when the imbalance is severe, the max-margin classifier essentially ignores the samples in the minority class and

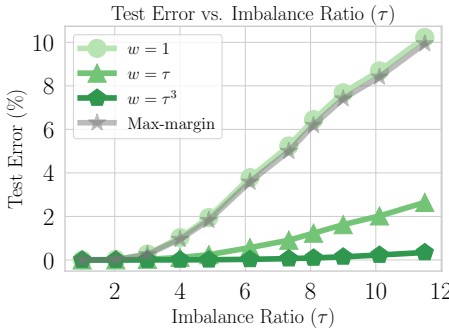 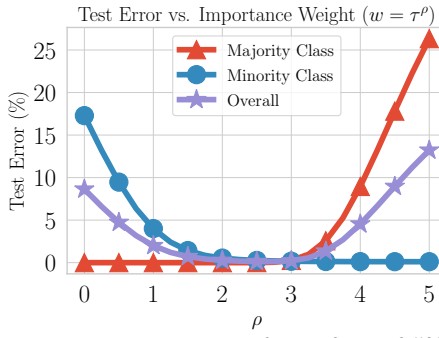

Figure 2: A simulation study with class-conditional Gaussians, with $d = 10^6$, $\|\mu\|^2 = d^{0.502}$ and $n = 100$. The mean $\mu_1 = \|\mu\|e_1$ and $\mu_2 = \|\mu\|e_2$, and $q \sim \mathsf{N}(0, I_{d \times d})$. (Left) We plot the test error of $\widehat{\theta}_1$ as $\tau$ varies for different choices of $w$, and also for the maximum margin classifier. The choice $w = \tau^3$ leads to the lowest error throughout, while $w = \tau$ also does well for small $\tau$. Both the polynomially-tailed classifier with no importance weighting and the maximum margin classifier suffer large test error. (Right) Here we fix the imbalance ratio $\tau = 10.1$, and study how the error of $\widehat{\theta}_1$ varies with $w$. When $w = \tau^3$, the error on both the majority and minority classes is almost equal, resulting in low overall test error.

overfits to the majority class. Ignoring these negative samples results in a relatively small alignment between the maximum margin classifier and the negative of the minority class center, which leads to high error on this class.

Together, these two theorems demonstrate that there is a strict gap between the performance of exponentially tailed and polynomially tailed losses under distribution shifts. As a concrete example, consider the scaling where $\|\mu\|^2 = d^{\frac{1}{2} + \frac{1}{40}}$, $n = d^{\frac{1}{5}}$ and $\tau = d^{\frac{3}{20}} \leq n$. For all large enough $d$, all our assumptions are satisfied. Now since, $\sqrt{n}\|\mu\|^2 > \sqrt{\tau d}$, Theorem 4.2 guarantees that the test error of $\widehat{\theta}_1 \to 0$ as $d \to \infty$. However, as $\tau = d^{\frac{3}{20}} \geq c'\|\mu\|^2\sqrt{n/d} = c'd^{\frac{5}{40}}$, for all large enough $d$, the test error of the maximum margin classifier is guaranteed to be at least $1/8$. This degradation of the test error with $\tau$ is also apparent in our simulations in Figure 2.

## 5 EMPIRICAL EVALUATION ON DEEP INTERPOLATING CLASSIFIERS

Inspired by our theoretical results, we use a polynomially-tailed loss with importance weights to train interpolating deep neural networks under distribution shift. We use a polynomially-tailed loss with $\beta = 1$ and $\ell_{\mathsf{left}}(z) = \log(1 + e^{-z})/\log(1 + e^{-1})$, ensuring that $\ell_{\alpha,\beta}$ is continuous at transition $z = \beta$ when $\alpha = 1$. We train models on two image classification datasets, one with label shift and one with subpopulation shift, and include additional experiment details in Appendix E. Though these nonlinear networks violate the assumptions of our theory, polynomially-tailed loss with importance weights consistently improves test accuracy for interpolating neural networks under distribution shift compared to importance-weighted cross-entropy loss.

**Imbalanced binary CIFAR10.** We construct our label shift dataset from the full CIFAR10 dataset. Similar to Byrd & Lipton (2019), we create a binary classification dataset out of the "cat" and "dog" classes. We use the official test examples as our label-balanced test set of 1000 cats and 1000 dogs. To form the train and validation sets, we use all 5000 cat examples but only 500 dog examples from the official train set, corresponding to a 10:1 label imbalance. We then use 80% of those examples for training and the rest for validation. We use the same convolutional neural network architecture as Byrd & Lipton (2019) with random initializations for this dataset.

**Subsampled CelebA.** For our subpopulation shift dataset, we use the CelebA with spurious correlations dataset constructed by Sagawa et al. (2019). This dataset has two class labels, "blonde hair" and "dark hair". Distinguishing examples by the "male" versus "female" attribute results in four total subpopulations, or groups. The distribution shift in the dataset comes from the change in relative proportions among the groups between the train and test sets. To reduce computation, we train on 2% of the full CelebA training set, resulting in group sizes of 1446, 1308, 468, and 33. We construct our test set by downsampling the original test to get group-balanced representation in

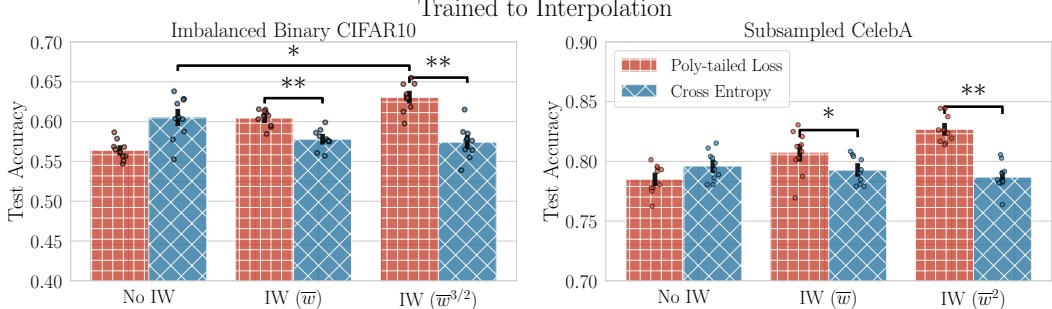

Figure 3: Polynomially-tailed loss versus cross-entropy loss on a label shift dataset and a subpopulation shift dataset for neural networks optimized past 100% train accuracy without regularization. $*$ and $**$ indicate $p < 0.05$ and $p < 0.005$ statistical significance, respectively. Importance weights (IW) consistently leads to gains when used with the polynomially-tailed loss across both datasets. Exponentiating the weights further amplifies these gains for the polynomially-tailed loss. IW has little effect for cross-entropy.[2]

the test set, resulting in 180 examples for each group. Following Sagawa et al. (2019), we use a ResNet-50 with ImageNet initialization for this dataset.

## 5.1 ISOLATING THE EFFECTS OF IMPORTANCE WEIGHTS

**Interpolating regime.** To understand the impact of polynomial losses in the interpolating regime, we train unregularized neural networks with SGD past 100% train accuracy and report their final accuracies on the test sets. We compare models trained with our polynomially-tailed loss against those trained with the cross-entropy loss which has exponential tails. Let $\overline{w}$ correspond to the unbiased importance weights. We consider three weighting scenarios for both loss functions: no importance weighting at all (**No IW**) the classical unbiased importance weighting (**IW** $(\overline{w})$), and biased weighting where we exponentiate the weights, (**IW** $(\overline{w}^c)$), increasing the ratio of different weights. The third setting is inspired by our theory in Section 4 that shows biased importance weights can improve performance for polynomially-tailed losses. For these experiments, we fixed $\alpha = 1$ and set the exponents to be the largest value that still allowed for stable optimization. For CelebA, we exponentiate by 2 and for CIFAR10 we exponentiate by $3/2$. We run 10 seeds for each experiment setting. We compare the two losses via paired one-sided Welch's $t$-tests, pairing runs with the same random seed. We report the exact numbers in Appendix E.

Figure 3 shows the mean accuracy and the standard error for each of the three settings. As indicated by our theory, we find that importance weighting with polynomially-tailed losses leads to statistically significant gains over cross-entropy in all cases. Further we find that exponentiating weights boosts the performance of polynomially tailed losses, confirming our claims in Theorem 4.2. However, exponentiating the weights leaves the performance of cross-entropy loss largely unchanged. In the case of CelebA, we find that exponentiated reweighting with poly-tailed losses outperforms all other methods. While in the case of Imbalanced Binary CIFAR10, we find a smaller gap between polynomially-tailed losses and cross-entropy, partially due to a substantially higher run-to-run variability in training. This variability does not affect our main conclusion: exponentiated reweighting with poly-tailed losses still outperforms cross-entropy with significance $p = 0.01$.

In Appendix D, we also compare the performance of polynomially-tailed losses with the cross-entropy loss when we regularize training via early-stopping. We find that even with regularization, poly-tailed losses give test accuracies better than or similar to cross-entropy across scenarios.

## 5.2 COMPARING AGAINST PRIOR DISTRIBUTION SHIFT METHODS

To place our method's empirical performance in the context of prior works, we extensively compare our method to state-of-the-art distribution shift correction methods. On binary CIFAR10, we com-

---

[2] The mean test accuracy for **No IW** with cross-entropy on CIFAR10 appears to deviate from **IW**, but this is due to the high variance across multiple runs of the model. The high variance arises due to 10:1 imbalanced training data, combined with the fact that we train the models until they interpolate the training data. These results for **No IW** differ slightly from those in Byrd & Lipton (2019) which used balanced training data and imbalanced test data.

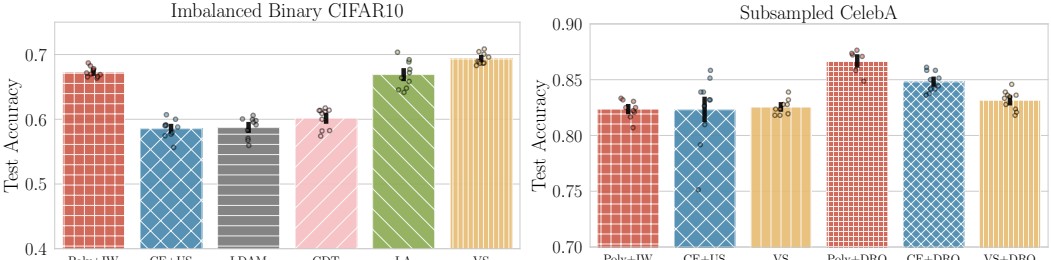

Figure 4: Polynomially-tailed loss with exponentiated importance weights is better than or competitive with state of the art methods that address distribution shift. We tune all methods extensively using the same validation set. We compare to cross entropy with undersampling (CE+US), label-distribution-aware margin loss (LDAM), class-dependent temperatures loss (CDT), logit-adjusted loss (LA), vector-scaling loss (VS), and the use of distributionally robust optimization (DRO).

pare to label-distribution-aware margin (LDAM) loss (Cao et al., 2019), class-dependent temperatures (CDT) loss (Ye et al., 2020), logit-adjusted (LA) loss (Menon et al., 2020), and vector-scaling (VS) loss (Kini et al., 2021). On CelebA, we compare to VS loss only since it encapsulates CDT loss, LA loss, and LDAM loss. We also evaluate the effects of distributionally robust optimization (DRO) by Sagawa et al. (2019) when combined with poly-tailed loss versus VS loss. We grid search each method exhaustively, and report results over 10 random initialization seeds. See Appendix E for more details on our procedure and the best hyperparameters we found. Unlike in Section 5.1, here we tune both the importance weight exponent *and* $\alpha$ for our poly-tailed loss. For all methods, including ones that use DRO, we train all models to interpolation without early stopping or strong regularizations, since we are interested in interpolating classifiers.

Figure 4 shows that when $\alpha$ and the weights' exponent are properly tuned, *our poly-tailed loss with exponentiated weights gives accuracies comparable to or even exceeding those by these recently proposed methods*. For both datasets, we found that cross entropy (CE) loss with undersampling (US), which discards data from overrepresented groups or classes to form a balanced training set, was a strong baseline.

On binary CIFAR10, our poly-tailed loss performs comparably to LA loss and is only 2% worse than VS loss in mean test accuracy with overlapping confidence intervals. At first, we grid searched VS loss across the published hyperparameter ranges from Kini et al. (2021), and it performed poorly (58.6% test accuracy). More thorough grid search for VS loss gave optimal hyperparameters that were similar to those of LA loss, increasing test accuracy by 10%. In sum, our poly-tailed loss matched or exceeded existing data imbalance adjustment methods (under published hyperparameters) and was beaten by 2% only after more extensively tuning the baselines.

On CelebA, our poly-tailed loss with IW achieves comparable test accuracy to VS loss. When we use both loss functions with DRO, poly loss with DRO is better than VS loss with DRO by 3.5% on test accuracy and 8% better on worst group accuracy. Poly-tailed loss with IW gives the best worst group accuracy out of all loss functions when not using DRO (see Appendix E). Such improved performance is surprising, as we designed polynomially tailed losses to optimize for static importance weights, rather than the worst case weights in DRO. The finding that poly-tailed losses with IW performs well on worst-group accuracy may be an interesting direction for future work

## 6 CONCLUSION

Contrary to popular belief, importance weights are in fact compatible with the current paradigm of overparameterized classifiers, *provided that the loss function does not decay exponentially*. The limiting behavior of an interpolating linear classifier trained with a weighted polynomially-tailed loss provably depends on the importance weights, unlike the maximum-margin classifier recovered by an exponentially decaying loss function. Our theoretical analysis of generalization error further suggested that exponentiating the classic importance weights can be key to accurate distribution shift correction. We empirically corroborated these theoretical intuitions, finding that weighted polynomially-tailed losses are also able to address distribution shift for interpolating neural networks. Our work suggests that heavy-tailed losses together with importance weighting serve as a simple and general candidate for addressing distribution shift in deep learning.

## ETHICS STATEMENT

We use publicly available datasets in our paper, and do not foresee any potential harm arising from the use of our methods.

## REPRODUCIBILITY STATEMENT

The full proofs of our theoretical results are presented in the appendix. All experimental details and hyperparameters are presented in Appendix E.

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

## A  PROOF OF PROPOSITION 4.1

First, we restate the statement of the proposition.

**Proposition 4.1.** *Suppose that the data is linearly separable. For any $\alpha > 0$, $\beta \in \mathbb{R}$, any initial point $\theta^{(0)} \in \mathbb{R}^d$, and for all small enough step-sizes $\eta$ the direction of the gradient descent iterates satisfy the following:* $\lim_{t\to\infty} \frac{\theta^{(t)}}{\|\theta^{(t)}\|} \to \widehat{\theta}_\alpha$.

**Proof** Recall that the loss $\ell_{\alpha,\beta}$ is assumed to be convex, strictly decreasing to zero and is differentiable. Define the minimizer of the loss over a ball of radius $R$ as follows

$$\theta_R = \operatorname*{arg\,min}_{\theta:\|\theta\|\leq R} \widehat{L}_{\alpha,\beta}(\theta).$$

Under the assumptions of this proposition, we can invoke Theorem 1 by Ji et al. (2020) to get that

$$\lim_{t\to\infty} \frac{\theta^{(t)}}{\|\theta^{(t)}\|} = \lim_{R\to\infty} \frac{\theta_R}{R}.$$

We will therefore instead demonstrate that $\lim_{R\to\infty} \frac{\theta_R}{R} = \widehat{\theta}_\alpha$ to establish our claim.

First, note that the classifier $\theta_R$ is the minimizer of loss in a ball of radius $R$, so there exists a $R_0$ with

$$\widehat{L}_{\alpha,\beta}(\theta_{R_0}) = \sum_{i\in[n]} w_i \ell_{\alpha,\beta}(z_i \cdot \theta_{R_0}) \leq (\min_i w_i)\ell_{\alpha,\beta}(\beta)$$

such that for all $R > R_0$, each example is classified correctly by $\theta_R$ and the margin $(z_i \cdot \theta_R)$ on each example is at least $\beta$. Therefore,

$$\widehat{L}_{\alpha,\beta}(\theta_R) = \sum_{i=1}^{n} w_i \ell_{\alpha,\beta}(z_i \cdot \theta_R) = \sum_{i=1}^{n} \frac{w_i}{[z_i \cdot \theta_R - (\beta-1)]^\alpha}.$$

Thus, for any radius $R > R_0$ we have

$$\frac{\theta_R}{R} = \frac{1}{R} \operatorname*{arg\,min}_{\theta:\|\theta\|\leq R} \widehat{L}_{\alpha,\beta}(\theta)$$

$$= \frac{1}{R} \operatorname*{arg\,min}_{\theta:\|\theta\|\leq R} \left\{ \sum_{i=1}^{n} \frac{w_i}{[z_i \cdot \theta - (\beta-1)]^\alpha}, \quad \text{s.t., } z_i \cdot \theta \geq \beta \right\}$$

$$= \frac{1}{R} \operatorname*{arg\,min}_{\theta:\|\theta\|\leq R} \left\{ \frac{1}{R^\alpha} \sum_{i=1}^{n} \frac{w_i}{\left[z_i \cdot \left(\frac{\theta}{R}\right) - \frac{\beta-1}{R}\right]^\alpha}, \quad \text{s.t., } z_i \cdot \left(\frac{\theta}{R}\right) \geq \frac{\beta}{R} \right\}$$

$$= \frac{1}{R} \times \left( R \cdot \operatorname*{arg\,min}_{\theta:\|\theta\|\leq 1} \left\{ \sum_{i=1}^{n} \frac{w_i}{\left[z_i \cdot \theta - \frac{\beta-1}{R}\right]^\alpha}, \quad \text{s.t., } z_i \cdot \theta \geq \frac{\beta}{R} \right\} \right).$$

Now taking the limit $R \to \infty$ we get that

$$\lim_{R\to\infty} \frac{\theta_R}{R} = \lim_{R\to\infty} \operatorname*{arg\,min}_{\theta:\|\theta\|\leq 1} \left\{ \sum_{i=1}^{n} \frac{w_i}{\left[z_i \cdot \theta - \frac{\beta-1}{R}\right]^\alpha}, \quad \text{s.t., } z_i \cdot \theta \geq \frac{\beta}{R} \right\}$$

$$\overset{(i)}{=} \operatorname*{arg\,min}_{\theta:\|\theta\|\leq 1} \left\{ \lim_{R\to\infty} \sum_{i=1}^{n} \frac{w_i}{\left[z_i \cdot \theta - \frac{\beta-1}{R}\right]^\alpha}, \quad \text{s.t., } z_i \cdot \theta \geq \frac{\beta}{R} \right\}$$

$$= \operatorname*{arg\,min}_{\theta:\|\theta\|\leq 1} \left\{ \sum_{i=1}^{n} \frac{w_i}{(z_i \cdot \theta)^\alpha}, \quad \text{s.t., } z_i \cdot \theta \geq 0 \right\}$$

$$\overset{(ii)}{=} \operatorname*{arg\,min}_{\theta:\|\theta\|\leq 1} \left\{ \sum_{i=1}^{n} \frac{w_i}{(z_i \cdot \theta)^\alpha}, \quad \text{s.t., } z_i \cdot \theta > 0 \right\}$$

$$= \widehat{\theta}_\alpha.$$

where $(i)$ follows since for every $R$ the function

$$\sum_{i=1}^{n} \frac{w_i}{\left[z_i \cdot \theta - \frac{\beta-1}{R}\right]^{\alpha}}$$

is convex and continuous, and therefore

$$\underset{\theta: \|\theta\| \leq 1}{\arg\min} \left\{\sum_{i=1}^{n} \frac{w_i}{\left[z_i \cdot \theta - \frac{\beta-1}{R}\right]^{\alpha}}, \quad \text{s.t., } z_i \cdot \theta \geq \frac{\beta}{R}\right\}$$

has a unique minimizer (since the objective function is strictly convex, and the constraint set is convex and compact), and is a continuous map by Berge's maximum theorem (see, e.g., Beavis & Dobbs, 1990, Theorem 3.6). Thus, it is possible to switch the order of the limit and the $\arg\min$. Equation $(ii)$ follows since the data is linearly separable, so there exists a $\theta$ such that $z_i \cdot \theta > 0$ for all $i$ which has finite objective value.

Therefore,

$$\lim_{t \to \infty} \frac{\theta^{(t)}}{\|\theta^{(t)}\|} = \lim_{R \to \infty} \frac{\theta_R}{R} = \widehat{\theta}_\alpha,$$

which finishes our proof. ∎

# B  PROOF OF THEOREM 4.2

In this section, we will prove an upper bound on the test error $\widehat{\theta}_1$, which is the classifier learnt by gradient descent with the importance weighted Polynomially-tailed loss that decays as $1/z$. By Proposition 4.1, we know the choice of $\ell_{\text{left}}$ and $\beta$ does not affect the implicit bias of gradient descent, hence here, for the sake of convenience, we shall use the specific loss function

$$\ell_{1,\beta}(z) := \begin{cases} -(z - \beta) + 1 & \text{if } z < \beta \\ \frac{1}{z - (\beta - 1)} & \text{if } z \geq \beta, \end{cases}$$

with $\beta = 1 - wn < 0$. It can be easily checked that this loss function is convex, differentiable and monotonically decreasing.

We want to bound the test error of $\widehat{\theta}_1$, however, in light of Proposition 4.1 we will instead bound the test error of the limiting iterate of gradient descent starting from $\theta^{(0)} = 8w^{2/3}n\left(\mu_1 - w^{1/3}\mu_2\right)$ with step-size $\eta > 0$ (again by the implicit bias of gradient descent is not affected by the choice of the initial point):

$$\theta^{(t+1)} = \theta^{(t)} - \eta\nabla\widehat{L}(\theta^{(t)}),$$

where

$$\widehat{L}(\theta) = \sum_{i \in \mathcal{P}} \ell_{1,\beta}(z_i \cdot \theta) + w\sum_{i \in \mathcal{N}} \ell_{1,\beta}(z_i \cdot \theta).$$

Recall that if the step-size is small enough then Proposition 4.1 guarantees that $\lim_{t \to \infty} \frac{\theta^{(t)}}{\|\theta^{(t)}\|} = \widehat{\theta}_1$. In this section we will simply let $\ell$ denote $\ell_{1,\beta}$ and also use the shorthands $\ell_i^{(t)} := \ell(z_i \cdot \theta^{(t)})$ and $\ell_i'^{(t)} := \ell'(z_i \cdot \theta^{(t)})$.

With this setup in place let us begin the proof. All of the assumptions made in Section 4.2.1 are in scope here. First we have a lemma that upper bounds the test error using Hoeffding's inequality.

**Lemma B.1.** *There is a positive absolute constant $c$ such that*

$$\mathbb{P}_{(x,y) \sim \mathsf{P}_{\text{test}}}\left[\text{sign}\left(\theta \cdot x\right) \neq y\right] \leq \frac{1}{2}\left[\exp\left(-c\frac{(\theta \cdot \mu_1)^2}{\|\theta\|^2}\right) + \exp\left(-c\frac{(\theta \cdot \mu_2)^2}{\|\theta\|^2}\right)\right].$$

This lemma follows by mirroring the proof of (Chatterji & Long, 2021, Lemma 9) which in turn follows by a simple application of Hoeffding's inequality (Vershynin, 2018, Theorem 2.6.3).

Next we have a lemma that proves bounds on the norms of the samples, bounds the inner products between the samples and also shows that with high probability the data is linearly separable.

Recall that the constants $C \geq 1$ and $0 < \widetilde{C} \leq 1$ were defined above in Section 4.2.

**Lemma B.2.** *For all $0 < \widetilde{C} \leq 1$, there is a $c \geq 1$ such that, for all large enough $C$, with probability $1 - \delta$ over the draw of the samples the following events simultaneously occur:*

1. *For all $k \in [n]$*
$$\frac{d}{c} \leq \|z_k\|^2 \leq cd.$$

2. *For all $i \in \mathcal{P}$ and $j \in \mathcal{N}$,*
$$|z_i \cdot z_j| \leq c\sqrt{d \log(n/\delta)}.$$

3. *For all $i \neq j \in [n]$,*
$$|z_i \cdot z_j| < c(\|\mu\|^2 + \sqrt{d \log(n/\delta)}).$$

4. *For all $k \in \mathcal{P}$,*
$$|\mu_1 \cdot z_k - \|\mu\|^2| < \|\mu\|^2/2.$$

5. *For all $k \in \mathcal{N}$,*
$$|(-\mu_2) \cdot z_k - \|\mu\|^2| < \|\mu\|^2/2.$$

6. *For all $k \in \mathcal{N}$,*
$$|\mu_1 \cdot z_k| < c\|\mu\|\sqrt{\log(n/\delta)}.$$

7. *For all $k \in \mathcal{P}$,*
$$|\mu_2 \cdot z_k| < c\|\mu\|\sqrt{\log(n/\delta)}.$$

8. *The samples are linearly separable.*

**Proof** We shall prove that each of the ten parts hold with probability at least $1 - \delta/10$ and take a union bound to prove our lemma. Under the assumptions of this lemma, Parts (1), (3)-(5) and (8) can be shown to hold with the required probability by invoking (Chatterji & Long, 2021, Lemma 13). We will show that Parts (2), (6) and (7) also hold with the required probability to complete the proof.

*Proof of Part* (2): Note that for all $i \in \mathcal{P}$, $z_i = \mu_1 + Uq_i$, and for all $i \in \mathcal{N}$, $z_i = \mu_2 + Uq_2$. Therefore, for any $i \in \mathcal{P}$ and $j \in \mathcal{N}$, since $\mu_1 \cdot \mu_2 = 0$,
$$
\begin{aligned}
|z_i \cdot z_j| &= \mu_1 \cdot (Uq_j) + \mu_2 \cdot (Uq_i) + q_i \cdot q_j \\
&\leq |\mu_1 \cdot (Uq_j)| + |\mu_2 \cdot (Uq_i)| + |q_i \cdot q_j| \\
&= |(U^\top \mu_1) \cdot q_j| + |(U^\top \mu_2) \cdot q_i| + |q_i \cdot q_j|. \quad (3)
\end{aligned}
$$
We will show that each of these terms is small with high probability for all pairs $i \in \mathcal{P}$ and $j \in \mathcal{N}$.

For the first two terms, note that by Hoeffding's inequality (Vershynin, 2018, Theorem 2.6.3)
$$
\begin{aligned}
\mathbb{P}\left[|(U^\top \mu_1) \cdot q_j| > \frac{c\|\mu\|\sqrt{\log(n/\delta)}}{3}\right] &< 2\exp\left(-\frac{c_1 c^3}{9} \frac{\|\mu\|^2 \log(n/\delta)}{\|\mu\|^2}\right) \\
&= 2\exp\left(-\frac{c_1 c^3}{9} \log(n/\delta)\right) \\
&= 2\left(\frac{\delta}{n}\right)^{c_1 c^3/9} \\
&\leq \frac{\delta}{30n},
\end{aligned}
$$

where the last inequality follows since the constant $c \geq 1$ can be chosen to be large enough and because $\delta < 1/C$ for a large enough constant $C$. Therefore, by taking a union bound over all $j \in \mathcal{N}$

$$\mathbb{P}\left[\exists j \in \mathcal{N}, \ |(U^\top \mu_1) \cdot q_j| > \frac{c\|\mu\|\sqrt{\log(n/\delta)}}{3}\right] < \frac{\delta}{30}. \tag{4}$$

Using an analogous argument we can also show that

$$\mathbb{P}\left[\exists i \in \mathcal{P}, \ |(U^\top \mu_2) \cdot q_i| > \frac{c\|\mu\|\sqrt{\log(n/\delta)}}{3}\right] < \frac{\delta}{30}. \tag{5}$$

Next, using (Chatterji & Long, 2021, inequality (15)) we get that

$$\mathbb{P}\left[\exists i \neq j \in [n], \ |q_i \cdot q_j| > \frac{c\sqrt{d\log(n/\delta)}}{3}\right] \leq \frac{\delta}{30}. \tag{6}$$

Combining inequalities (3)-(6) we get that for all $i \in \mathcal{P}$ and $j \in \mathcal{N}$ with probability at least $1 - \delta/10$

$$|z_i \cdot z_j| \leq \frac{2c\|\mu\|\sqrt{\log(n/\delta)}}{3} + \frac{c\sqrt{d\log(n/\delta)}}{3} \leq c\sqrt{d\log(n/\delta)},$$

where the last inequality follows since by assumption $d \geq Cn\|\mu\|^2$ and because $C$ is large enough. This completes the proof of this part.

*Proof of Part* (6)*:* For any $k \in \mathcal{N}$, $z_k = \mu_2 + Uq_k$. Recall that $\mu_1 \cdot \mu_2 = 0$, thus

$$|\mu_1 \cdot z_k| = |\mu_1 \cdot (Uq_k)|.$$

Invoking inequality (4) proves that this part holds with probability at least $1 - \delta/10$.

*Proof of Part* (7)*:* This follows by an analogous argument as in the previous part. ∎

We continue by defining a good event that we will work under for the rest of the proof.

**Definition B.3.** *If the training dataset $\mathcal{S}$ satisfies all the conditions specified in Lemma B.2 then we call it a* good run*.*

Going forward in this section we shall assume that a good run occurs.

The following lemma provides some useful bound on the loss ratio at initialization and guarantees that the loss of example remains in the polynomially-tailed part throughout training.

**Lemma B.4.** *Recall that $\theta^{(0)} = 8w^{2/3}n(\mu_1 - w^{1/3}\mu_2)$, and that $w_i = 1$ if $i \in \mathcal{P}$ and $w_i = w$ if $i \in \mathcal{N}$. On a good run, for all $i \neq j \in [n]$*

$$\frac{\ell_i^{(0)}}{\ell_j^{(0)}} \leq 8\left(\frac{w_j}{w_i}\right)^{1/3}.$$

*Furthermore, if the step-size $\eta$ is sufficiently small then on a good run, for all $t \in \{0, 1, \ldots\}$ and for all $i \in [n]$*

$$\ell_i^{(t)} = \frac{1}{z_i \cdot \theta^{(t)} - (\beta - 1)}.$$

**Proof** Consider an $i \in \mathcal{P}$ then

$$z_i \cdot \theta^{(0)} = 8w^{2/3}n\left(z_i \cdot \mu_1 - w^{1/3}z_i \cdot \mu_2\right) \overset{(i)}{\leq} 8w^{2/3}n\left(\frac{3\|\mu\|^2}{2} + c_1 w^{1/3}\|\mu\|\sqrt{\log(n/\delta)}\right)$$

$$\leq 12w^{2/3}n\|\mu\|^2\left[1 + \frac{2c_1 w^{1/3}\sqrt{\log(n/\delta)}}{3\|\mu\|}\right]$$

$$\overset{(ii)}{\leq} 16w^{2/3}n\|\mu\|^2,$$

where $(i)$ follows by Lemma B.2 and $(ii)$ follows since $\|\mu\|^2 \geq Cn^2 \log(n/\delta) \geq C\tau^2 \log(n/\delta) \geq \frac{C}{2^{2/3}} w^{2/3} \log(n/\delta)$ for a large enough constant $C$. Similarly, we also have the lower bound

$$z_i \cdot \theta^{(0)} = 8w^{2/3}n \left( z_i \cdot \mu_1 - w^{1/3} z_i \cdot \mu_2 \right)$$

$$\geq 8w^{2/3}n \left( \frac{\|\mu\|^2}{2} - c_1 w^{1/3}\|\mu\|\sqrt{\log(n/\delta)} \right) \geq 4w^{2/3}n\|\mu\|^2 \left[ 1 - \frac{2c_1 w^{1/3}\sqrt{\log(n/\delta)}}{\|\mu\|} \right].$$

Recall from the definition of the loss that we set $\beta = 1 - nw$. Now again since $\|\mu\|^2 \geq \frac{C}{2^{2/3}} w^{2/3} \log(n/\delta)$, and because we choose $\beta = 1 - nw$, we infer that

$$z_i \cdot \theta^{(0)} - (\beta - 1) \geq \frac{8w^{2/3}n\|\mu\|^2}{3} - nw \geq 2w^{2/3}n\|\mu\|^2.$$

Since the margin of the point is larger than $(\beta - 1)$ therefore by the definition of the loss function above we have that

$$\ell_i^{(0)} = \frac{1}{z_i \cdot \theta^{(0)} - (\beta - 1)}$$

and hence

$$\frac{1}{16w^{2/3}n\|\mu\|^2} \leq \ell_i^{(0)} \leq \frac{1}{2w^{2/3}n\|\mu\|^2}. \tag{7}$$

By mirroring the logic we can also prove that for all $j \in \mathcal{N}$

$$\frac{1}{16wn\|\mu\|^2} \leq \ell_j^{(0)} \leq \frac{1}{2wn\|\mu\|^2}. \tag{8}$$

By combining equations (7) and (8) we immediately get that

$$\frac{\ell_i^{(0)}}{\ell_j^{(0)}} \leq 8 \left( \frac{w_j}{w_i} \right)^{1/3}.$$

This proves the first part of the lemma.

To prove the second part we shall prove that the loss on each example remains smaller than $1/2$, which ensures $\ell_i^{(t)}$ is equal to the polynomial loss function. Note that

$$\widehat{L}(\theta^{(0)}) = \sum_{i \in \mathcal{P}} \ell_i^{(0)} + w \sum_{i \in \mathcal{N}} \ell_i^{(0)} \leq \frac{|\mathcal{P}|}{2w^{2/3}n\|\mu\|^2} + \frac{w|\mathcal{N}|}{2wn\|\mu\|^2} \leq \frac{wn}{2wn\|\mu\|^2} \leq 1/2. \tag{9}$$

This proves that $\ell_i^{(0)} \leq 1/2$ for all $i \in [n]$. By (Ji et al., 2020, Lemma 2) we know that if the step-size is small enough then the sequence $\{\widehat{L}(\theta^{(t)})\}$ is non-increasing. Hence, we have that $\ell_i^{(t)} \leq 1/2$ for all $t \in \{0, 1, \ldots\}$ and for all samples. This wraps up our proof. ∎

The next lemma lower bounds the inner product between the normalized gradient descent iterates, and $\mu_1$ and $\mu_2$.

**Lemma B.5.** *Let $c$ and $c'$ be absolute constants. Then, on a good run, for any $t \in \{0, 1, \ldots\}$*

$$\frac{\mu_1 \cdot \theta^{(t+1)}}{\|\theta^{(t+1)}\|} \geq \frac{\mu_1 \cdot \theta^{(0)}}{\|\theta^{(t+1)}\|} + \frac{\|\mu\|^2\sqrt{|\mathcal{N}|}}{2c\sqrt{d}} \left[ \frac{1}{1 + \frac{\|\theta^{(0)}\|}{c\eta\sqrt{\frac{d}{|\mathcal{N}|}}\sum_{s=0}^{t}\sum_{i \in [n]} -w_i \ell_i'^{(s)}}} \right] \times$$

$$\frac{\sum_{s=0}^{t} \left[ \sum_{i \in \mathcal{P}} -w_i \ell_i'^{(s)} - \frac{c'\sqrt{\log(n/\delta)}}{\|\mu\|} \sum_{i \in \mathcal{N}} -w_i \ell_i'^{(s)} \right]}{\sum_{s=0}^{t}\sum_{i \in [n]} -w_i \ell_i'^{(s)}}$$

*and*

$$\frac{\mu_2 \cdot \theta^{(t+1)}}{\|\theta^{(t+1)}\|} \geq \frac{\mu_2 \cdot \theta^{(0)}}{\|\theta^{(t+1)}\|} + \frac{\|\mu\|^2\sqrt{|\mathcal{N}|}}{2c\sqrt{d}} \left[ \frac{1}{1 + \frac{\|\theta^{(0)}\|}{c\eta\sqrt{\frac{d}{|\mathcal{N}|}}\sum_{s=0}^{t}\sum_{i \in [n]} -w_i \ell_i'^{(s)}}} \right] \times$$

$$\frac{\sum_{s=0}^{t} \left[ \sum_{i \in \mathcal{N}} -w_i \ell_i'^{(s)} - \frac{c'\sqrt{\log(n/\delta)}}{\|\mu\|} \sum_{i \in \mathcal{P}} -w_i \ell_i'^{(s)} \right]}{\sum_{s=0}^{t}\sum_{i \in [n]} -w_i \ell_i'^{(s)}}.$$

**Proof** Let us prove the first claim for the inner product with $\mu_1$. The second claim regarding $\mu_2$ shall follow analogously. For any $t \in \{0, 1, \ldots\}$, by the definition of the gradient descent step, we have that

$$\mu_1 \cdot \theta^{(t+1)} = \mu_1 \cdot \theta^{(t)} + \eta \sum_{i \in \mathcal{P}} w_i \mu_1 \cdot z_i \left(-\ell_i'^{(t)}\right) + \eta \sum_{i \in \mathcal{N}} w_i \mu_1 \cdot z_i \left(-\ell_i'^{(t)}\right).$$

Note that by the definition of the loss function we have that $-\ell_i'^{(t)} \geq 0$ for all $t$ and all $i$. Thus by Parts (1) and (6) of Lemma B.2 we get that

$$\mu_1 \cdot \theta^{(t+1)} \geq \mu_1 \cdot \theta^{(t)} + \frac{\eta \|\mu\|^2}{2} \sum_{i \in \mathcal{P}} -w_i \ell_i'^{(t)} - c_1 \eta \|\mu\| \sqrt{\log(n/\delta)} \sum_{i \in \mathcal{N}} -w_i \ell_i'^{(t)}$$

$$= \mu_1 \cdot \theta^{(t)} + \frac{\eta \|\mu\|^2}{2} \sum_{i \in [n]} -w_i \ell_i'^{(t)} - \frac{\eta \|\mu\|^2}{2} \left(1 + \frac{c_1 \sqrt{\log(n/\delta)}}{\|\mu\|}\right) \sum_{i \in \mathcal{N}} -w_i \ell_i'^{(t)}$$

$$= \mu_1 \cdot \theta^{(t)} + \frac{\eta \|\mu\|^2}{2} \left[\sum_{i \in \mathcal{P}} -w_i \ell_i'^{(t)} - \frac{c_1 \sqrt{\log(n/\delta)}}{\|\mu\|} \sum_{i \in \mathcal{N}} -w_i \ell_i'^{(t)}\right],$$

where $c_1$ is the constant from Lemma B.2. Unrolling this inequality over $t$ steps we have that

$$\mu_1 \cdot \theta^{(t+1)} = \mu_1 \cdot \theta^{(0)} + \frac{\eta \|\mu\|^2}{2} \sum_{s=0}^{t} \left[\sum_{i \in \mathcal{P}} -w_i \ell_i'^{(s)} - \frac{c_1 \sqrt{\log(n/\delta)}}{\|\mu\|} \sum_{i \in \mathcal{N}} -w_i \ell_i'^{(s)}\right]. \tag{10}$$

On the other hand, by the triangle inequality we know that

$$\|\theta^{(t+1)}\| \leq \|\theta^{(t)}\| + \eta \|\nabla \widehat{L}(\theta^{(t)})\|$$

$$= \|\theta^{(t)}\| + \eta \left\|\sum_{i \in [n]} z_i(-w_i \ell_i'^{(t)})\right\|$$

$$\leq \|\theta^{(t)}\| + \eta \left\|\sum_{i \in \mathcal{P}} z_i(-\ell_i'^{(t)})\right\| + \eta w \left\|\sum_{i \in \mathcal{N}} z_i(-\ell_i'^{(t)})\right\|. \tag{11}$$

Now note that

$$\left\|\sum_{i \in \mathcal{P}} z_i(-\ell_i'^{(t)})\right\|^2 = \sum_{i \in \mathcal{P}} (\ell_i'^{(t)})^2 \|z_i\|^2 + \sum_{i \neq j \in \mathcal{P}} \ell_i'^{(t)} \ell_j'^{(t)} (z_i \cdot z_j)$$

$$\overset{(i)}{\leq} c_1 \left[\sum_{i \in \mathcal{P}} (-\ell_i'^{(t)})^2 d + \sum_{i \neq j \in \mathcal{P}} \ell_i'^{(t)} \ell_j'^{(t)} (\|\mu\|^2 + \sqrt{d \log(n/\delta)})\right]$$

$$\overset{(ii)}{\leq} c_1 \left(\max_{j \in \mathcal{P}} -\ell_j'^{(t)}\right) \left[d \sum_{i \in \mathcal{P}} -\ell_i'^{(t)} + |\mathcal{P}|(\|\mu\|^2 + \sqrt{d \log(n/\delta)}) \sum_{i \in \mathcal{P}} -\ell_i'^{(t)}\right]$$

$$= c_1 \left(\max_{j \in \mathcal{P}} -\ell_j'^{(t)}\right) (d + |\mathcal{P}|\|\mu\|^2 + |\mathcal{P}|\sqrt{d \log(n/\delta)}) \sum_{i \in \mathcal{P}} -\ell_i'^{(t)}$$

$$\overset{(iii)}{\leq} c_2 \frac{d}{|\mathcal{P}|} \left(\sum_{i \in \mathcal{P}} -\ell_i'^{(t)}\right)^2$$

where $(i)$ follows by using Part (1) and Part (3) of Lemma B.2, $(ii)$ follows since $-\ell_j'^{(t)}$ is positive for all $j \in [n]$, and $(iii)$ follows since by assumption $d \geq Cn\|\mu\|^2$ and $\|\mu\|^2 \geq Cn^2 \log(n/\delta)$ for a large enough constant $C$. Hence we have that

$$\left\|\sum_{i \in \mathcal{P}} z_i(-\ell_i'^{(t)})\right\| \leq \sqrt{c_2 \frac{d}{|\mathcal{P}|}} \sum_{i \in \mathcal{P}} -\ell_i'^{(t)}.$$

Similarly, one can also show that

$$\left\| \sum_{i \in \mathcal{N}} z_i (-\ell_i'^{(t)}) \right\| \leq \sqrt{c_2 \frac{d}{|\mathcal{N}|} \sum_{i \in \mathcal{N}} -\ell_i'^{(t)}}.$$

Applying these bounds in inequality (11) we get that

$$\|\theta^{(t+1)}\| \leq \|\theta^{(t)}\| + \eta \left[ \sqrt{c_2 \frac{d}{|\mathcal{P}|} \sum_{i \in \mathcal{P}} -\ell_i'^{(t)}} + w \sqrt{c_2 \frac{d}{|\mathcal{N}|} \sum_{i \in \mathcal{N}} -\ell_i'^{(t)}} \right]$$

$$\leq \|\theta^{(t)}\| + \eta \left[ \sqrt{c_2 \frac{d}{|\mathcal{N}|} \sum_{i \in \mathcal{P}} -\ell_i'^{(t)}} + w \sqrt{c_2 \frac{d}{|\mathcal{N}|} \sum_{i \in \mathcal{N}} -\ell_i'^{(t)}} \right]$$

$$= \|\theta^{(t)}\| + c_3 \eta \sqrt{\frac{d}{|\mathcal{N}|} \sum_{i \in [n]} -w_i \ell_i'^{(t)}}.$$

Therefore, unrolling this bound over $t$ steps we find that

$$\|\theta^{(t+1)}\| \leq \|\theta^{(0)}\| + c_3 \eta \sqrt{\frac{d}{|\mathcal{N}|}} \sum_{s=0}^{t} \sum_{i \in [n]} -w_i \ell_i'^{(s)} \tag{12}$$

$$= c_3 \eta \sqrt{\frac{d}{|\mathcal{N}|}} \sum_{s=0}^{t} \sum_{i \in [n]} -w_i \ell_i'^{(s)} \left[ 1 + \frac{\|\theta^{(0)}\|}{c_3 \eta \sqrt{\frac{d}{|\mathcal{N}|}} \sum_{s=0}^{t} \sum_{i \in [n]} -w_i \ell_i'^{(s)}} \right].$$

Thus, combined with inequality (10) we get that,

$$\frac{\mu_1 \cdot \theta^{(t+1)}}{\|\theta^{(t+1)}\|} \geq \frac{\mu_1 \cdot \theta^{(0)}}{\|\theta^{(t+1)}\|} + \frac{\|\mu\|^2 \sqrt{|\mathcal{N}|}}{2 c_3 \sqrt{d}} \left[ \frac{1}{1 + \frac{\|\theta^{(0)}\|}{c_3 \eta \sqrt{\frac{d}{|\mathcal{N}|}} \sum_{s=0}^{t} \sum_{i \in [n]} -w_i \ell_i'^{(s)}}} \right] \times$$

$$\frac{\sum_{s=0}^{t} \left[ \sum_{i \in \mathcal{P}} -w_i \ell_i'^{(s)} - \frac{c_1 \sqrt{\log(n/\delta)}}{\|\mu\|} \sum_{i \in \mathcal{N}} -w_i \ell_i'^{(s)} \right]}{\sum_{s=0}^{t} \sum_{i \in [n]} -w_i \ell_i'^{(s)}},$$

which completes the proof of the first part of the lemma. The second part follows by an identical argument. ∎

Next, we prove a lemma that shows that throughout training the ratio between the losses between any two samples remains bounded.

**Lemma B.6.** *There is a positive absolute constant $c \geq 1$ such that the following holds for all large enough $C$, and all small enough step-sizes $\eta$ and for any $\frac{\tau^3}{2} \leq w \leq 2\tau^3$. On a good run, for all $t \in \{1, 2, \ldots\}$ and all $i \neq j \in [n]$*

$$\frac{\ell_i^{(t)}}{\ell_j^{(t)}} \leq c \left( \frac{w_j}{w_i} \right)^{1/3}.$$

**Proof** Let $c_1 \geq 1$ be the constant $c$ from Lemma B.2 above. We shall show that the choice $c = \max\{8, 2(4c_1^2)^{1/3}\}$ suffices.

We shall prove this via an inductive argument. For the base case, at step $t = 0$, we know that by Lemma B.4 that the ratio between the losses of sample $i$ and $j$ is upper bounded by $8 (w_j/w_i)^{1/3}$. Now, we shall assume that the inductive hypothesis holds at an arbitrary step $t > 0$ and prove that it holds at step $t + 1$.

Without loss of generality, we shall analyze the ratio between the losses of the samples with indices $i = 1$ and $j = 2$. A similar analysis shall hold for any other pair. Define $G_t := \ell_1^{(t)}$, $H_t := \ell_2^{(t)}$,

$A_t := H_t/G_t$. Note that,

$$
\begin{aligned}
G_{t+1} &= \frac{1}{\theta^{(t+1)} \cdot z_1 - (\beta - 1)} \\
&= \frac{1}{\theta^{(t)} \cdot z_1 + \eta \sum_{k \in [n]} w_k(-\ell_k'^{(t)})(z_k \cdot z_1) - (\beta - 1)} \\
&\overset{(i)}{=} \frac{1}{\theta^{(t)} \cdot z_1 - (\beta - 1) + \eta \sum_{k \in [n]} w_k(z_k \cdot z_1)\left(\ell_k^{(t)}\right)^2} \\
&= \frac{\ell_1^{(t)}}{1 + \eta \ell_1^{(t)} \sum_{k \in [n]} w_k(z_k \cdot z_1)\left(\ell_k^{(t)}\right)^2} \\
&= G_t \cdot \frac{1}{1 + \eta \ell_1^{(t)} \sum_{k \in [n]} w_k(z_k \cdot z_1)\left(\ell_k^{(t)}\right)^2}
\end{aligned}
$$

where $(i)$ follows since $-\ell_k'^{(t)} = (\ell_k^{(t)})^2$ as the loss of each example is always in the polynomial tail of the loss by Lemma B.4. Therefore, we have that

$$
\begin{aligned}
A_{t+1} = \frac{H_{t+1}}{G_{t+1}} &= \frac{H_t}{G_t} \cdot \frac{1 + \eta \ell_1^{(t)} \sum_{k \in [n]} w_k(z_k \cdot z_1)\left(\ell_k^{(t)}\right)^2}{1 + \eta \ell_2^{(t)} \sum_{k \in [n]} w_k(z_k \cdot z_2)\left(\ell_k^{(t)}\right)^2} \\
&= A_t \cdot \frac{1 + \eta \ell_1^{(t)} \sum_{k \in [n]} w_k(z_k \cdot z_1)\left(\ell_k^{(t)}\right)^2}{1 + \eta \ell_2^{(t)} \sum_{k \in [n]} w_k(z_k \cdot z_2)\left(\ell_k^{(t)}\right)^2}.
\end{aligned}
$$

Now since the step-size $\eta$ is chosen to be small enough, $|z_i \cdot z_j| \le c_1 d$ (by Part (**??**) of Lemma B.2 and by the assumption on $d$) and because the losses are all smaller than a constant by Lemma B.4, the following approximations hold

$$
1 + \eta \ell_1^{(t)} \sum_{k \in [n]} w_k(z_k \cdot z_1)\left(\ell_k^{(t)}\right)^2 \le \exp\left(\eta \ell_1^{(t)} \sum_{k \in [n]} w_k(z_k \cdot z_1)\left(\ell_k^{(t)}\right)^2\right)
$$

$$
1 + \eta \ell_2^{(t)} \sum_{k \in [n]} w_k(z_k \cdot z_2)\left(\ell_k^{(t)}\right)^2 \ge \exp\left(\frac{\eta \ell_2^{(t)}}{2} \sum_{k \in [n]} w_k(z_k \cdot z_2)\left(\ell_k^{(t)}\right)^2\right),
$$

and thus,

$$
A_{t+1} \le A_t \exp\left(\eta \ell_1^{(t)} \sum_{k \in [n]} w_k(z_k \cdot z_1)\left(\ell_k^{(t)}\right)^2 - \frac{\eta \ell_2^{(t)}}{2} \sum_{k \in [n]} w_k(z_k \cdot z_2)\left(\ell_k^{(t)}\right)^2\right). \tag{13}
$$

Let us further upper bound the RHS as follows

$$
\begin{aligned}
&A_t \exp\left(\eta \ell_1^{(t)} \sum_{k \in [n]} w_k(z_k \cdot z_1)\left(\ell_k^{(t)}\right)^2 - \frac{\eta \ell_2^{(t)}}{2} \sum_{k \in [n]} w_k(z_k \cdot z_2)\left(\ell_k^{(t)}\right)^2\right) \\
&= A_t \exp\left(\eta w_1 \left(\ell_1^{(t)}\right)^3 \|z_1\|^2 - \frac{1}{2}\eta w_2 \left(\ell_2^{(t)}\right)^3 \|z_2\|^2\right) \\
&\qquad \times \exp\left(\eta \ell_1^{(t)} \sum_{k \ne 1} w_k(z_k \cdot z_1)\left(\ell_k^{(t)}\right)^2 - \frac{1}{2}\eta \ell_2^{(t)} \sum_{k \ne 2} w_k(z_k \cdot z_2)\left(\ell_k^{(t)}\right)^2\right) \\
&\overset{(i)}{\le} A_t \exp\left(c_1 \eta d w_1 \left(\ell_1^{(t)}\right)^3 - \frac{1}{2c_1}\eta d w_2 \left(\ell_2^{(t)}\right)^3\right) \\
&\qquad \times \exp\left(c_2 \eta \left(\ell_1^{(t)} + \frac{\ell_2^{(t)}}{2}\right)\left(\|\mu\|^2 + \sqrt{d \log(n/\delta)}\right) \sum_{k \in [n]} w_k \left(\ell_k^{(t)}\right)^2\right)
\end{aligned}
$$

where $(i)$ follows since for all $i \in [n]$, $d/c_1 \le \|z_i\|^2 \le c_1 d$ and for any $j \ne k$, $|z_j \cdot z_k| \le c_1 \left( \|\mu\|^2 + \sqrt{d \log(n/\delta)} \right)$ by Lemma B.2. Continuing we get that,

$$
\begin{aligned}
A_{t+t} &\le A_t \exp \left( c_1 \eta d w_1 G_t^3 - \frac{1}{2c_1} \eta d w_2 H_t^3 \right) \\
&\quad \times \exp \left( c_2 \eta \left( G_t + H_t \right) \left( \|\mu\|^2 + \sqrt{d \log(n/\delta)} \right) \sum_{k \in [n]} w_k \left( \ell_k^{(t)} \right)^2 \right) \\
&= A_t \exp \left( -\frac{1}{2c_1} \eta d w_2 G_t^3 \left( A_t^3 - \frac{2c_1^2 w_1}{w_2} \right) \right) \\
&\quad \times \exp \left( c_2 \eta \left( G_t + H_t \right) \left( \|\mu\|^2 + \sqrt{d \log(n/\delta)} \right) \sum_{k \in [n]} w_k \left( \ell_k^{(t)} \right)^2 \right),
\end{aligned}
\tag{14}
$$

With this upper bound in place, consider two cases.

**Case 1** ($A_t^3 \le \frac{4c_1^2 w_1}{w_2}$)**:** Using inequality (14) we know that

$$
\begin{aligned}
A_{t+1} &\le A_t \exp \left( -\frac{1}{2c_1} \eta d w_2 G_t^3 \left( A_t^3 - \frac{2c_1^2 w_1}{w_2} \right) \right) \\
&\quad \times \exp \left( c_2 \eta \left( G_t + H_t \right) \left( \|\mu\|^2 + \sqrt{d \log(n/\delta)} \right) \sum_{k \in [n]} w_k \left( \ell_k^{(t)} \right)^2 \right) \\
&\le A_t \exp \left( c_1 \eta d w_1 G_t^3 \right) \exp \left( c_2 \eta \left( G_t + H_t \right) \left( \|\mu\|^2 + \sqrt{d \log(n/\delta)} \right) \sum_{k \in [n]} w_k \left( \ell_k^{(t)} \right)^2 \right).
\end{aligned}
$$

Now the loss on each example is less than the total initial loss $1/2$ (see equation (9)) and therefore,

$$
\begin{aligned}
A_{t+1} &\le A_t \exp \left( c_3 \eta d w \right) \exp \left( c_4 \eta \left( \|\mu\|^2 + \sqrt{d \log(n/\delta)} \right) w n \right) \\
&\overset{(i)}{\le} \left( \frac{4c_1^2 w_1}{w_2} \right)^{1/3} \exp(1/8) \le 2 \left( \frac{4c_1^2 w_1}{w_2} \right)^{1/3} \overset{(ii)}{\le} c \left( \frac{w_1}{w_2} \right)^{1/3},
\end{aligned}
$$

where $(i)$ follows by choosing the step-size $\eta$ to be small enough and because $A_t \le \left( \frac{4c_1^2 w_1}{w_2} \right)^{1/3}$ in this case, and $(ii)$ follows by the choice of constant $c \ge 2(4c_1^2)^{1/3}$ from above.

**Case 2** ($\frac{4c_1^2 w_1}{w_2} < A_t^3 \le \frac{c^3 w_1}{w_2}$)**:** In this case again by inequality (14)

$$
\begin{aligned}
A_{t+1} &\le A_t \exp \left( -\frac{1}{2c_1} \eta d w_2 G_t^3 \left( A_t^3 - \frac{2c_1^2 w_1}{w_2} \right) \right) \\
&\quad \times \exp \left( c_2 \eta \left( G_t + H_t \right) \left( \|\mu\|^2 + \sqrt{d \log(n/\delta)} \right) \sum_{k \in [n]} w_k \left( \ell_k^{(t)} \right)^2 \right) \\
&= A_t \exp \left( -\frac{1}{2c_1} \eta d w_2 G_t^3 \left( A_t^3 - \frac{2c_1^2 w_1}{w_2} \right) \right) \\
&\quad \times \exp \left( c_2 \eta G_t^3 \left( A_t + 1 \right) \left( \|\mu\|^2 + \sqrt{d \log(n/\delta)} \right) \sum_{k \in [n]} w_k \frac{\left( \ell_k^{(t)} \right)^2}{G_t^2} \right).
\end{aligned}
$$

Continuing we find that,

$$A_{t+1} \overset{(i)}{\leq} A_t \exp\left(-\frac{1}{2c_1}\eta dw_2 G_t^3 \left(A_t^3 - \frac{2c_1^2 w_1}{w_2}\right)\right)$$

$$\times \exp\left(c_6 \eta G_t^3 \max\left\{\left(\frac{w_1}{w_2}\right)^{1/3}, 1\right\}\left(\|\mu\|^2 + \sqrt{d\log(n/\delta)}\right)\sum_{k\in[n]} w_k \left(\frac{w_1}{w_k}\right)^{2/3}\right)$$

$$= A_t \exp\left(-\frac{1}{2c_1}\eta dw_2 G_t^3 \left(A_t^3 - \frac{2c_1^2 w_1}{w_2}\right)\right)$$

$$\times \exp\left(c_6 \eta w_1 G_t^3 \left(\|\mu\|^2 + \sqrt{d\log(n/\delta)}\right)\sum_{k\in[n]} \left(\frac{w_k}{\min\{w_1, w_2\}}\right)^{1/3}\right)$$

$$\overset{(ii)}{\leq} A_t \exp\left(-\frac{1}{2c_1}\eta dw_2 G_t^3 \left(A_t^3 - \frac{2c_1^2 w_1}{w_2}\right)\right)$$

$$\times \exp\left(c_6 \eta w_1 G_t^3 \left(\|\mu\|^2 + \sqrt{d\log(n/\delta)}\right)\left(|\mathcal{P}| + w^{1/3}|\mathcal{N}|\right)\right)$$

$$= A_t \exp\left(-\frac{1}{2c_1}\eta dw_2 G_t^3 \left(A_t^3 - \frac{2c_1^2 w_1}{w_2}\right)\right)$$

$$\times \exp\left(c_6 \eta w_1 G_t^3 \left(\|\mu\|^2 + \sqrt{d\log(n/\delta)}\right)|\mathcal{P}|\left(1 + \frac{w^{1/3}}{\tau}\right)\right)$$

$$\overset{(iii)}{\leq} A_t \exp\left(-\frac{1}{2c_1}\eta dw_2 G_t^3 \left(A_t^3 - \frac{2c_1^2 w_1}{w_2}\right)\right)\exp\left(2c_6\eta w_1 G_t^3 \left(\|\mu\|^2 + \sqrt{d\log(n/\delta)}\right)n\right)$$

$$\overset{(iv)}{\leq} A_t \exp\left(-\frac{1}{2c_1}\eta dw_2 G_t^3 \cdot \frac{2c_1^2 w_1}{w_2}\right)\exp\left(2c_6\eta w_1 G_t^3 \left(\|\mu\|^2 + \sqrt{d\log(n/\delta)}\right)n\right)$$

$$= A_t \exp\left[-c_1 w_1 G_t^3 \left(d - c_7 n\left(\|\mu\|^2 + \sqrt{d\log(n/\delta)}\right)\right)\right],$$

where $(i)$ follows by the inductive hypothesis which guarantees that $\frac{\ell_k^{(t)}}{\ell_1^{(t)}} \leq c(\frac{w_1}{w_k})^{1/3} \leq cw^{1/3}$, $(ii)$ follows since $w_k = 1$ if $k \in \mathcal{P}$ and $w_k = w$ if $k \in \mathcal{N}$. Inequality $(iii)$ follows since $w < 2\tau^3$ and since $|\mathcal{P}| \leq n$, and finally $(iv)$ follows since in this case $A_t^3 > \frac{4c_1^2 w_1}{w_2}$. Now since by assumption the dimension

$$d \geq Cn\|\mu\|^2 \geq C^2 n^3 \log(n/\delta)$$

for a large enough constant $C$. Hence, we have that $A_{t+1} \leq A_t \leq c\left(\frac{w_1}{w_2}\right)^{1/3}$ in this case.

Recall that since we assumed the inductive hypothesis to hold, the two cases analyzed above are exhaustive. This completes our proof. ∎

The next lemma uses the loss ratio bound that we established to show that the difference between the gradient of the losses over the positive cluster and the negative cluster is small at any iteration.

**Lemma B.7.** *For any positive $\tilde{c}$, there exists a $0 < c < 1$ such that, for all large enough $C$, if the step-size $\eta$ is sufficiently small and $\frac{\tau^3}{2} \leq w \leq 2\tau^3$ then on a good run, for any $t \in \{0, 1, \ldots\}$*

$$\sum_{i\in\mathcal{P}} -w_i \ell_i'^{(t)} - \frac{\tilde{c}\sqrt{\log(n/\delta)}}{\|\mu\|}\sum_{i\in\mathcal{N}} -w_i \ell_i'^{(t)} \geq c\sum_{i\in[n]} -w_i \ell_i'^{(t)}$$

*and*

$$\sum_{i\in\mathcal{N}} -w_i \ell_i'^{(t)} - \frac{\tilde{c}\sqrt{\log(n/\delta)}}{\|\mu\|}\sum_{i\in\mathcal{P}} -w_i \ell_i'^{(t)} \geq c\sum_{i\in[n]} -w_i \ell_i'^{(t)}.$$

**Proof** We begin by proving the first part of the lemma. Note that since $[n] = \mathcal{P} \cup \mathcal{N}$ to prove the first part it suffices to instead show that

$$(1-c)\sum_{i\in\mathcal{P}} -w_i\ell_i'^{(t)} \geq \left(c + \frac{\tilde{c}\sqrt{\log(n/\delta)}}{\|\mu\|}\right)\sum_{i\in\mathcal{N}} -w_i\ell_i'^{(t)}$$

$$\iff (1-c)\sum_{i\in\mathcal{P}} -\ell_i'^{(t)} \geq \left(c + \frac{\tilde{c}\sqrt{\log(n/\delta)}}{\|\mu\|}\right)w\sum_{i\in\mathcal{N}} -\ell_i'^{(t)}$$

$$\iff (1-c)\sum_{i\in\mathcal{P}} (\ell_i^{(t)})^2 \geq \left(c + \frac{\tilde{c}\sqrt{\log(n/\delta)}}{\|\mu\|}\right)w\sum_{i\in\mathcal{N}} (\ell_i^{(t)})^2$$

$$\text{(since } -\ell_i'^{(t)} = (\ell_i^{(t)})^2 \text{ for the polynomial loss)}$$

$$\Leftarrow (1-c)|\mathcal{P}|\min_{i\in\mathcal{P}}(\ell_i^{(t)})^2 \geq \left(c + \frac{\tilde{c}\sqrt{\log(n/\delta)}}{\|\mu\|}\right)w|\mathcal{N}|\max_{i\in\mathcal{N}}(\ell_i^{(t)})^2$$

$$\Leftarrow (1-c)\tau \geq \left(c + \frac{\tilde{c}\sqrt{\log(n/\delta)}}{\|\mu\|}\right)w\frac{\max_{i\in\mathcal{N}}(\ell_i^{(t)})^2}{\min_{i\in\mathcal{P}}(\ell_i^{(t)})^2}$$

$$\Leftarrow (1-c)\tau \geq \left(c + \frac{\tilde{c}\sqrt{\log(n/\delta)}}{\|\mu\|}\right)wc_1 \cdot \frac{1}{w^{2/3}}$$

$$\text{(by invoking Lemma B.6; note that } c_1 \geq 1)$$

$$\Leftarrow \frac{(1-c)}{c_1\left(c + \frac{\tilde{c}}{\sqrt{C}}\right)}\tau \geq w^{1/3}$$

$$\text{(since } \|\mu\| \geq \sqrt{Cn^2\log(n/\delta)}, \text{ where } C \text{ is a large enough constant)}.$$

Since $C$ is large enough, we now choose the constant $0 < c < 1$ to be such that $(1-c)/(c_1(c + \tilde{c}/\sqrt{C}))$ is at least $(2)^{1/3}$. This proves the first part of the lemma.

To prove the second part of the lemma, note that again since $[n] = \mathcal{P} \cup \mathcal{N}$ it suffices to show that

$$(1-c)\sum_{i\in\mathcal{N}} -w_i\ell_i'^{(t)} \geq \left(c + \frac{\tilde{c}\sqrt{\log(n/\delta)}}{\|\mu\|}\right)\sum_{i\in\mathcal{P}} -w_i\ell_i'^{(t)}$$

$$\Leftarrow (1-c)|\mathcal{N}|w\min_{i\in\mathcal{N}}(\ell_i^{(t)})^2 \geq \left(c + \frac{\tilde{c}\sqrt{\log(n/\delta)}}{\|\mu\|}\right)|\mathcal{P}|\max_{i\in\mathcal{P}}(\ell_i^{(t)})^2$$

$$\Leftarrow w \geq \frac{\left(c + \frac{\tilde{c}\sqrt{\log(n/\delta)}}{\|\mu\|}\right)}{(1-c)}\frac{\max_{i\in\mathcal{P}}(\ell_i^{(t)})^2}{\min_{i\in\mathcal{N}}(\ell_i^{(t)})^2} \cdot \tau$$

$$\Leftarrow w \geq \frac{\left(c + \frac{\tilde{c}\sqrt{\log(n/\delta)}}{\|\mu\|}\right)}{(1-c)}c_1w^{2/3}\tau$$

$$\Leftarrow w^{1/3} \geq \frac{c_1\left(c + \frac{\tilde{c}}{\sqrt{C}}\right)}{(1-c)} \cdot \tau$$

$$\Leftarrow w^{1/3} \geq \frac{\tau}{2^{1/3}}$$

where the last implication follows by the choice of $c$ from above. This completes the proof. ∎

We now have all the pieces required to prove our main theorem. Recall its statement.

**Theorem 4.2.** *For any $0 < \widetilde{C} \leq 1$, there is a constant $c$ such that, for all large enough $C$ and for any $0 < \delta < 1/C$, under the assumptions of this subsection the following holds. If the weight*

$\frac{\tau^3}{2} \leq w \leq 2\tau^3$, then with probability at least $1 - \delta$, training on $\mathcal{S}$ produces a classifier $\widehat{\theta}_1$ satisfying:

$$\mathsf{TestError}[\widehat{\theta}_1] = \mathbb{P}_{(x,y)\sim\mathsf{P}_{\mathsf{test}}}\left[\mathrm{sign}\left(\widehat{\theta}_1 \cdot x\right) \neq y\right] \leq \exp\left(-\frac{cn}{\tau}\frac{\|\mu\|^4}{d}\right).$$

**Proof** First, by Part (8) of Lemma B.2 we know that the data is linearly separable. Thus, by Proposition 4.1 we know that

$$\widehat{\theta}_1 = \lim_{t\to\infty}\frac{\theta^{(t)}}{\|\theta^{(t)}\|}.$$

Given this equivalence, by Lemma B.1 we know that

$$\mathbb{P}_{(x,y)\sim\mathsf{P}_{\mathsf{test}}}\left[\mathrm{sign}\left(\widehat{\theta}_1 \cdot x\right) \neq y\right]$$

$$= \mathbb{P}_{(x,y)\sim\mathsf{P}_{\mathsf{test}}}\left[\mathrm{sign}\left(\lim_{t\to\infty}\frac{\theta^{(t)}}{\|\theta^{(t)}\|} \cdot x\right) \neq y\right]$$

$$\leq \frac{1}{2}\left[\exp\left(-c'\left(\lim_{t\to\infty}\frac{\theta^{(t)}\cdot\mu_1}{\|\theta^{(t)}\|}\right)^2\right) + \exp\left(-c'\left(\lim_{t\to\infty}\frac{\theta^{(t)}\cdot\mu_2}{\|\theta^{(t)}\|}\right)^2\right)\right]. \qquad (15)$$

We shall now establish lower bounds on $\lim_{t\to\infty}\frac{\theta^{(t)}\cdot\mu_1}{\|\theta^{(t)}\|}$ and $\lim_{t\to\infty}\frac{\theta^{(t)}\cdot\mu_2}{\|\theta^{(t)}\|}$ to obtain the desired bound on the test error. Let us lower bound $\lim_{t\to\infty}\frac{\theta^{(t)}\cdot\mu_1}{\|\theta^{(t)}\|}$. The bound on $\lim_{t\to\infty}\frac{\theta^{(t)}\cdot\mu_2}{\|\theta^{(t)}\|}$ shall follow by exactly the same logic.

By Lemma B.5 we know that

$$\lim_{t\to\infty}\frac{\mu_1\cdot\theta^{(t+1)}}{\|\theta^{(t+1)}\|}$$

$$\geq \lim_{t\to\infty}\frac{\mu_1\cdot\theta^{(0)}}{\|\theta^{(t+1)}\|} + \lim_{t\to\infty}\frac{\|\mu\|^2\sqrt{|\mathcal{N}|}}{2c_1\sqrt{d}}\left[\frac{1}{1 + \frac{\|\theta^{(0)}\|}{c_1\eta\sqrt{\frac{d}{|\mathcal{N}|}}\sum_{s=0}^{t}\sum_{i\in[n]}-w_i\ell_i'^{(s)}}}\right] \times$$

$$\sum_{s=0}^{t}\left[\sum_{i\in\mathcal{P}}-w_i\ell_i'^{(s)} - \frac{c_2\sqrt{\log(n/\delta)}}{\|\mu\|}\sum_{i\in\mathcal{N}}-w_i\ell_i'^{(s)}\right]$$

$$\overset{(i)}{\geq} \frac{\|\mu\|^2\sqrt{|\mathcal{N}|}}{2c_1\sqrt{d}}\cdot\lim_{t\to\infty}\left[\frac{1}{1 + \frac{\|\theta^{(0)}\|}{c_1\eta\sqrt{\frac{d}{|\mathcal{N}|}}\sum_{s=0}^{t}\sum_{i\in[n]}-w_i\ell_i'^{(s)}}}\right] \times$$

$$\sum_{s=0}^{t}\left[\sum_{i\in\mathcal{P}}-w_i\ell_i'^{(s)} - \frac{c_2\sqrt{\log(n/\delta)}}{\|\mu\|}\sum_{i\in\mathcal{N}}-w_i\ell_i'^{(s)}\right]$$

$$= \frac{\|\mu\|^2\sqrt{|\mathcal{N}|}}{2c_1\sqrt{d}}\cdot\lim_{t\to\infty}\left[\frac{1}{1 + \frac{\|\theta^{(0)}\|}{c_1\eta\sqrt{\frac{d}{|\mathcal{N}|}}\sum_{s=0}^{t}\sum_{i\in[n]}-w_i\ell_i'^{(s)}}}\right] \times$$

$$\lim_{t\to\infty}\sum_{s=0}^{t}\left[\sum_{i\in\mathcal{P}}-w_i\ell_i'^{(s)} - \frac{c_2\sqrt{\log(n/\delta)}}{\|\mu\|}\sum_{i\in\mathcal{N}}-w_i\ell_i'^{(s)}\right], \qquad (16)$$

where $(i)$ follows since $\mu_1\cdot\theta^{(0)}$ is bounded and $\|\theta^{(t+1)}\| \to \infty$ by (Ji et al., 2020, Lemma 2).

We will now show that the first limit in RHS equals 1. To do this, first note that

$$1 \geq \lim_{t\to\infty}\left[\frac{1}{1 + \frac{\|\theta^{(0)}\|}{c_1\eta\sqrt{\frac{d}{|\mathcal{N}|}}\sum_{s=0}^{t}\sum_{i\in[n]}-w_i\ell_i'^{(s)}}}\right] = \left[\frac{1}{1 + \lim_{t\to\infty}\frac{\|\theta^{(0)}\|}{c_1\eta\sqrt{\frac{d}{|\mathcal{N}|}}\sum_{s=0}^{t}\sum_{i\in[n]}-w_i\ell_i'^{(s)}}}\right].$$

Now we will show that

$$\lim_{t \to \infty} \frac{\|\theta^{(0)}\|}{c_1 \eta \sqrt{\frac{d}{|\mathcal{N}|}} \sum_{s=0}^{t} \sum_{i \in [n]} -w_i \ell_i'^{(s)}} = 0.$$

First note that $\frac{\|\theta^{(0)}\|}{c_1 \eta \sqrt{d} \sum_{s=0}^{t} \sum_{i \in [n]} -w_i \ell_i'^{(s)}} \geq 0$, since $-w_i \ell_i'^{(s)} \geq 0$, so to show that this limit equals 0, it suffices to show that $c_1 \eta \sqrt{d} \sum_{s=0}^{t} \sum_{i \in [n]} -w_i \ell_i'^{(s)}$ grows unboundedly as $t \to \infty$. Using inequality (12) from above we get that,

$$\|\theta^{(t+1)}\| \leq \|\theta^{(0)}\| + c_3 \eta \sqrt{\frac{d}{|\mathcal{N}|}} \sum_{s=0}^{t} \sum_{i \in [n]} -w_i \ell_i'^{(s)}.$$

The norm $\|\theta^{(0)}\|$ is finite and we know that $\|\theta^{(t)}\| \to \infty$ by (Ji et al., 2020, Lemma 2), therefore $c_1 \eta \sqrt{d} \sum_{s=0}^{t} \sum_{i \in [n]} -w_i \ell_i'^{(s)}$ must grow unboundedly. This proves that

$$\lim_{t \to \infty} \left[ \frac{1}{1 + \frac{\|\theta^{(0)}\|}{c_1 \eta \sqrt{d} \sum_{s=0}^{t} \sum_{i \in [n]} -w_i \ell_i'^{(s)}}} \right] = 1$$

as claimed. This combined with inequality (16) yields the bound

$$\begin{aligned}
\lim_{t \to \infty} \frac{\mu_1 \cdot \theta^{(t+1)}}{\|\theta^{(t+1)}\|} &\geq \frac{\|\mu\|^2 \sqrt{|\mathcal{N}|}}{2c_1 \sqrt{d}} \lim_{t \to \infty} \frac{\sum_{s=0}^{t} \left[ \sum_{i \in \mathcal{P}} -w_i \ell_i'^{(s)} - \frac{c_2 \sqrt{\log(n/\delta)}}{\|\mu\|} \sum_{i \in \mathcal{N}} -w_i \ell_i'^{(s)} \right]}{\sum_{s=0}^{t} \sum_{i \in [n]} -w_i \ell_i'^{(s)}} \\
&\overset{(i)}{\geq} \frac{c_3 \|\mu\|^2 \sqrt{|\mathcal{N}|}}{\sqrt{d}} \lim_{t \to \infty} \frac{\sum_{s=0}^{t} \sum_{i \in [n]} -w_i \ell_i'^{(s)}}{\sum_{s=0}^{t} \sum_{i \in [n]} -w_i \ell_i'^{(s)}} \\
&= \frac{c_3 \|\mu\|^2 \sqrt{|\mathcal{N}|}}{\sqrt{d}} \\
&= \frac{c_3 \|\mu\|^2 \sqrt{|\mathcal{P}|}}{\sqrt{\tau d}} \\
&\overset{(ii)}{\geq} \frac{c_4 \|\mu\|^2 \sqrt{n}}{\sqrt{\tau d}}
\end{aligned}$$

where $(i)$ follows by invoking Lemma B.7 and $(ii)$ follows since $|\mathcal{P}| \geq n/2$. As stated above, using a similar argument we can also show that $\lim_{t \to \infty} \frac{\mu_1 \cdot \theta^{(t+1)}}{\|\theta^{(t+1)}\|} \geq \frac{c_4 \|\mu\|^2 \sqrt{n}}{\sqrt{\tau d}}$. Plugging these lower bounds into inequality (15) completes our proof. ∎

## C  PROOF OF THEOREM 4.3

In this section we will prove a lower bound on the test error of the maximum margin linear classifier $\widehat{\theta}_{\mathrm{MM}}$. Here we will work with the exponential loss

$$\ell(z) = \exp(-z).$$

Define the loss $\widehat{L}(\theta) = \sum_{i \in [n]} \ell(y_i x_i^\top \theta) = \sum_{i \in [n]} \ell(z_i^\top \theta)$. For a step-size $\eta$ and initial iterate $\theta^{(0)} = 0$ (this choice of initial point is for the sake of convenience, it does not affect the implicit bias of gradient descent) for any $t \in \{0, 1, \ldots\}$

$$\theta^{(t+1)} = \theta^{(t)} - \eta \nabla \widehat{L}(\theta^{(t)})$$

be the iterates of gradient descent. We let $\ell_i^{(t)}$ be shorthand for $\ell(z_i \cdot \theta^{(t)})$ and therefore,

$$\theta^{(t+1)} = \theta^{(t)} - \eta \nabla \widehat{L}(\theta^{(t)}) = \theta^{(t)} + \eta \sum_{i \in [n]} z_i \ell_i^{(t)}.$$

The results by Soudry et al. (2018) guarantee that for a small enough step-size $\eta$ the direction of the iterates of gradient descent $\lim_{t \to \infty} \frac{\theta^{(t)}}{\|\theta^{(t)}\|} = \widehat{\theta}_{\mathsf{MM}}$. Therefore, we will instead prove a lower bound on the asymptotic iterates of gradient descent.

As we did in the proof of Theorem 4.2, going forward we will assume that a good run occurs (see Definition B.3), which guarantees that all of the conditions specified in Lemma B.2 are satisfied by the training dataset $\mathcal{S}$.

With the setup in place, we shall now prove this theorem in stages. Throughout this section the assumptions stated in Section 4.2.1 shall remain in force. We begin with a lemma that shows that the margin of the maximum margin classifier scales with $\sqrt{d/n}$.

**Lemma C.1.** *There is an absolute constant $c$ such that, on a good run, for all large enough $C$, for all $i \in [n]$*

$$\widehat{\theta}_{\mathsf{MM}} \cdot z_i \geq c\sqrt{\frac{d}{n}}.$$

**Proof** We will prove this result by constructing a unit vector $\phi$ with a margin that scales with $\sqrt{d/n}$. This immediately implies that the maximum margin classifier must also attain this margin on all of the points.

Define $\phi$ to be as follows

$$\phi := \frac{\sum_{i \in [n]} z_i}{\|\sum_{j \in [n]} z_j\|}.$$

We will first bound the norm of the denominator as follows

$$
\begin{aligned}
\left\| \sum_{j \in [n]} z_j \right\|^2 &= \sum_{j \in [n]} \|z_j\|^2 + \sum_{j \neq k} z_j \cdot z_k \\
&\leq \sum_{j \in [n]} \|z_j\|^2 + \sum_{j \neq k} |z_j \cdot z_k| \\
&\overset{(i)}{\leq} c_1 nd + c_1 n^2 \left( \|\mu\|^2 + \sqrt{d \log(n/\delta)} \right) \\
&= c_1 nd \left( 1 + \frac{c_1 \|\mu\|^2}{d} + \frac{n\sqrt{\log(n/\delta)}}{\sqrt{d}} \right) \\
&\overset{(ii)}{\leq} 2 c_1 nd
\end{aligned}
\tag{17}
$$

where $(i)$ follows by Lemma B.2 and $(ii)$ follows since $d \geq Cn\|\mu\|^2 \geq C^2 n^3 \log(n/\delta)$ where recall $C$ is sufficiently large.

Now we lower bound the margin between numerator of $v$ and $z_k$ for any $k \in [n]$

$$
\begin{aligned}
\left( \sum_{i \in [n]} z_i \right) \cdot z_k = \|z_i\|^2 + \sum_{i \neq k} z_i \cdot z_k &\geq \|z_i\|^2 - \sum_{k \neq i} |z_i \cdot z_k| \\
&\overset{(i)}{\geq} \frac{d}{c_1} - c_1 n(\|\mu\|^2 + \sqrt{d \log(n/\delta)}) \\
&= \frac{d}{c_1} \left( 1 - \frac{c_1^2 n \|\mu\|^2}{d} - \frac{c_1^2 n \sqrt{\log(n/\delta)}}{\sqrt{d}} \right) \\
&\geq \frac{d}{2c_1},
\end{aligned}
\tag{18}
$$

where $(i)$ again follows by invoking Lemma B.2 and by the assumption on $d$,

Combining inequalities (17) and (18) yields that for any $k \in [n]$

$$\phi \cdot z_k \geq \frac{d}{2c_1\sqrt{2c_1 nd}} = c\sqrt{\frac{d}{n}}.$$

This proves the result.                                                                                 ∎

The next lemma provides control over the rate at which the norm of iterates $\theta^{(t)}$ grows late in training.

**Lemma C.2.** *There is an absolute constant $c$ such that, for all large enough $C$, if the step-size $\eta$ is sufficiently small then on a good run, there exists a $t_0$ such that for all $t \geq t_0$*

$$\|\theta^{(t+1)}\| \geq \|\theta^{(t)}\| + c\eta\sqrt{\frac{d}{n}}\sum_{i\in[n]}\ell_i^{(t)}.$$

**Proof** By the definition of gradient descent

$$\|\theta^{(t+1)}\|^2 = \|\theta^{(t)} + \eta\sum_{i\in[n]}\ell_i^{(t)}z_i\|^2$$

$$= \|\theta^{(t)}\|^2 + 2\eta\sum_{k\in[n]}\ell_i^{(t)}z_i \cdot \theta^{(t)} + \eta^2\|\sum_{i\in[n]}\ell_i^{(t)}z_i\|^2$$

$$\geq \|\theta^{(t)}\|^2 + 2\eta\sum_{i\in[n]}\ell_i^{(t)}z_i \cdot \theta^{(t)}. \tag{19}$$

Now we know that $\frac{\theta^{(t)}}{\|\theta^{(t)}\|} \to \widehat{\theta}_{\mathsf{MM}}$. Also note that in the previous lemma (Lemma C.1) we showed that for all $i \in [n]$

$$\widehat{\theta}_{\mathsf{MM}} \cdot z_i \geq c_1\sqrt{\frac{d}{n}}.$$

Therefore, there exists a iteration $t_1$ such that for all $t \geq t_1$ and all $i \in [n]$

$$\frac{\theta^{(t)} \cdot z_i}{\|\theta^{(t)}\|} \geq \frac{c_1}{2}\sqrt{\frac{d}{n}}.$$

Continuing from (19), for any $t \geq t_1$

$$\|\theta^{(t+1)}\|^2 \geq \|\theta^{(t)}\|^2 + 2\eta\sum_{i\in[n]}\ell_i^{(t)}z_i \cdot \theta^{(t)}$$

$$\geq \|\theta^{(t)}\|^2 + c_1\eta\|\theta^{(t)}\|\sqrt{\frac{d}{n}}\sum_{i\in[n]}\ell_i^{(t)}$$

$$= \|\theta^{(t)}\|^2\left[1 + \frac{c_1\eta\sqrt{\frac{d}{n}}\sum_{i\in[n]}\ell_i^{(t)}}{\|\theta^{(t)}\|}\right].$$

Taking square roots we get that for any $t \geq t_1$

$$\|\theta^{(t+1)}\| \geq \|\theta^{(t)}\|\sqrt{1 + \frac{c_1\eta\sqrt{\frac{d}{n}}\sum_{i\in[n]}\ell_i^{(t)}}{\|\theta^{(t)}\|}}. \tag{20}$$

Further by (Ji et al., 2020, Lemma 2) we know that $\|\theta^{(t)}\| \to \infty$, so there exists a $t_2$ such that for all $t \geq t_2$, $\|\theta^{(t)}\| \geq \frac{c_1\eta\sqrt{nd}}{8}$. Thus, for $t \geq t_2$, we have

$$\frac{1}{\|\theta^{(t)}\|} \cdot c_1\eta\sqrt{\frac{d}{n}}\sum_{i\in[n]}\ell_i^{(t)} \leq 8 \cdot \frac{\sum_{i\in[n]}\ell_i^{(t)}}{n} \leq 8, \tag{21}$$

where the last inequality follows since the initial loss is equal to $n$, as $\theta^{(0)} = 0$, and the total loss is decreasing again by (Ji et al., 2020, Lemma 2) if the step-size is small enough. It is easy to check that for any $0 \leq x \leq 8$

$$\sqrt{1+x} \geq 1 + \frac{x}{4}.$$

Thus, combining inequalities (20) and (21) we get that for all $t \geq t_0 = \max\{t_1, t_2\}$

$$\|\theta^{(t+1)}\| \geq \|\theta^{(t)}\|\sqrt{1 + \frac{c_1\eta\sqrt{\frac{d}{n}}\sum_{i \in [n]}\ell_i^{(t)}}{\|\theta^{(t)}\|}} \geq \|\theta^{(t)}\|\left(1 + \frac{c_1\eta\sqrt{\frac{d}{n}}\sum_{i \in [n]}\ell_i^{(t)}}{4\|\theta^{(t)}\|}\right)$$

$$= \|\theta^{(t)}\| + \frac{c_1}{4}\eta\sqrt{\frac{d}{n}}\sum_{i \in [n]}\ell_i^{(t)},$$

which completes our proof. ∎

Continuing we will show that throughout training the ratio of the losses between the different examples are bounded by a constant. This ensures that each example roughly "influences" the gradient update by the same amount in each step. However, since the number of points from the positive cluster is larger, the gradient update shall overall be more highly correlated with the mean of the majority positive center $\mu_1$ than the mean of the minority negative center $\mu_2$.

The proof is identical to the proof of Lemma 11 by Chatterji & Long (2021). However, since our setting is slightly different to the setting studied in that paper we reprove the result here.

**Lemma C.3.** *There is an absolute constant $c$ such that, for all large enough $C$, and all small enough step sizes $\eta$, on a good run, for all iterations $t \in \{0, 1, \ldots\}$ and all $i, j \in [n]$*

$$\frac{\ell_i^{(t)}}{\ell_j^{(t)}} \leq c.$$

**Proof** First note that $\widehat{L}(\theta^{(0)}) = \sum_{i \in [n]}\ell_i^{(0)} = n$ and since step-size $\eta$ is small enough training loss is non-increasing by (Ji et al., 2020, Lemma 2).

Let $c_1$ be the constant $c \geq 1$ from Lemma B.2. We will show that $c = 4c_1^2$ suffices.

We shall prove this via an inductive argument. For the base case, at step $t = 0$, we know that $\theta^{(0)} = 0$, therefore the loss on all of the samples is equal to 1. Now, we shall assume that the inductive hypothesis holds at an arbitrary step $t > 0$ and prove that it holds at step $t + 1$.

Without loss of generality, we shall analyze the ratio between the losses of the samples with indices $i = 1$ and $j = 2$. A similar analysis shall hold for any other pair. Define $G_t := \ell_1^{(t)}$, $H_t := \ell_2^{(t)}$, $A_t := H_t/G_t$. The ratio between these losses at step $t + 1$ is

$$A_{t+1} = \frac{\exp(-\theta^{(t+1)} \cdot z_2)}{\exp(-\theta^{(t+1)} \cdot z_1)}$$

$$= \frac{\exp(-(\theta^{(t)} + \eta\sum_{k \in [n]}\ell_k^{(t)}z_k) \cdot z_2)}{\exp(-(\theta^{(t)} + \eta\sum_{k \in [n]}\ell_k^{(t)}z_k) \cdot z_1)}$$

$$= A_t \cdot \exp\left(-\eta\sum_{k \in [n]}\ell_k^{(t)}(z_k \cdot z_2 - z_k \cdot z_1)\right)$$

$$= A_t \cdot \exp\left(-\eta\left(\ell_2^{(t)}\|z_2\|^2 - \ell_1^{(t)}\|z_1\|^2 - \sum_{k \neq 2}\ell_k^{(t)}z_k \cdot z_2 + \sum_{k \neq 1}\ell_k^{(t)}z_k \cdot z_1\right)\right)$$

$$= A_t \cdot \exp\left(-\eta\left(H_t\|z_2\|^2 - G_t\|z_1\|^2 - \sum_{k \neq 2}\ell_k^{(t)}z_k \cdot z_2 + \sum_{k \neq 1}\ell_k^{(t)}z_k \cdot z_1\right)\right)$$

$$\leq A_t \cdot \exp\left(-\eta\left(H_t\|z_2\|^2 - G_t\|z_1\|^2 - \sum_{k \neq 2}\ell_k^{(t)}|z_k \cdot z_2| - \sum_{k \neq 1}\ell_k^{(t)}|z_k \cdot z_1|\right)\right).$$

Now note that by Lemma B.2, for all $i \neq j \in [n]$, $d/c_1 \leq \|z_i\|^2 \leq c_1 d$ and $|z_i \cdot z_j| \leq c_1 \left( \|\mu\|^2 + \sqrt{d \log(n/\delta)} \right)$, and therefore,

$$
\begin{aligned}
A_{t+1} &\leq A_t \cdot \exp\left( -\eta \left( \frac{H_t d}{c_1} - G_t c_1 d - 2c_1 \left( \|\mu\|^2 + \sqrt{d \log(n/\delta)} \right) \sum_{k \in [n]} \ell_k^{(t)} \right) \right) \\
&= A_t \cdot \exp\left( -\eta \left( \frac{G_t d}{c_1} (A_t - c_1^2) - 2c_1 \left( \|\mu\|^2 + \sqrt{d \log(n/\delta)} \right) G_t \sum_{k \in [n]} \frac{\ell_k^{(t)}}{G_t} \right) \right) \\
&\overset{(i)}{\leq} A_t \cdot \exp\left( -\eta \left( \frac{G_t d}{c_1} (A_t - c_1^2) - 8c_1^3 \left( \|\mu\|^2 + \sqrt{d \log(n/\delta)} \right) G_t n \right) \right) \\
&= A_t \cdot \exp\left( -\frac{\eta G_t d}{c_1} \left( A_t - c_1^2 - 8c_1^4 \left( \frac{\|\mu\|^2 n}{d} + \frac{n\sqrt{\log(n/\delta)}}{\sqrt{d}} \right) \right) \right) \\
&\overset{(ii)}{\leq} A_t \cdot \exp\left( -\frac{\eta G_t d}{c_1} \left( A_t - c_1^2 - 8c_1^4 \left( \frac{1}{C} + \frac{1}{C} \right) \right) \right) \\
&\overset{(iii)}{\leq} A_t \cdot \exp\left( -\frac{\eta G_t d}{c_1} \left( A_t - 2c_1^2 \right) \right),
\end{aligned}
\tag{22}
$$

where $(i)$ follows since by the inductive hypothesis $\ell_k^{(t)}/G_t \leq c = 4c_1^2$, $(ii)$ follows since by assumption $d \geq Cn\|\mu\|^2 \geq C^2 n^3 \log(n/\delta)$, and $(iii)$ follows since $C$ is sufficiently large.

Now consider two cases.

**Case 1** ($A_t \leq 2c_1^2$)**:** By inequality (22)

$$
A_{t+1} \leq A_t \cdot \exp\left( -\frac{\eta G_t d}{c_1} (A_t - 2c_1^2) \right) = A_t \exp\left( \frac{\eta G_t d}{c_1} (2c_1^2 - A_t) \right) \begin{aligned} &\leq 2c_1^2 \exp\left( 2\eta c_1 G_t d \right) \\ &\leq 2c_1^2 \exp\left( 2\eta c_1 n d \right) \\ &\leq 4c_1^2, \end{aligned}
$$

where the last inequality follows if the step-size $\eta$ is sufficiently small.

**Case 2** ($2c_1^2 \leq A_t \leq 4c_1^2 = c$)**:** Again by inequality (22)

$$
A_{t+1} \leq A_t \cdot \exp\left( -\frac{\eta G_t d}{c_1} (A_t - 2c_1^2) \right) \leq A_t \leq 4c_1^2.
$$

Thus, we have shown that $A_{t+1} \leq 4c_1^2 = c$ in both cases. Since we assumed the induction hypothesis to hold at step $t$, these two cases are exhaustive, and hence the induction is complete. ∎

The next lemma proves an upper bound on the difference between the inner product between $\theta^{(t+1)} \cdot \mu_2$ and the corresponding inner product at iteration $t$. Since we start at $\theta^{(0)}$, unrolling this over $t$ steps gives us an upper bound on inner product $\theta^{(t)} \cdot \mu_2$ for any $t \geq 0$.

**Lemma C.4.** *There is an absolute constant $c$ such that, for all large enough $C$, if the step-size $\eta$ is sufficiently small then on a good run, for all $t \in \{0, 1, \ldots\}$*

$$
\left( \theta^{(t+1)} - \theta^{(t)} \right) \cdot (-\mu_2) \leq \frac{c\eta \|\mu\|^2}{\tau} \sum_{i \in [n]} \ell_i^{(t)}.
$$

**Proof** By the definition of a gradient descent step

$$
\begin{aligned}
(\theta^{(t+1)} - \theta^{(t)}) \cdot (-\mu_2) &= \eta \sum_{i \in [n]} \ell_i^{(t)} z_i \cdot (-\mu_2) \\
&\overset{(i)}{\leq} \frac{3\eta\|\mu\|^2}{2} \sum_{i \in \mathcal{N}} \ell_i^{(t)} + c_1 \eta \|\mu\| \sqrt{\log(n/\delta)} \sum_{i \in \mathcal{P}} \ell_i^{(t)} \\
&\overset{(ii)}{\leq} \frac{3\eta\|\mu\|^2}{2} \cdot \frac{c_2|\mathcal{N}|}{n} \sum_{i \in [n]} \ell_i^{(t)} + c_1 \eta \|\mu\| \sqrt{\log(n/\delta)} \cdot \frac{c_2|\mathcal{P}|}{n} \sum_{i \in [n]} \ell_i^{(t)} \\
&= \frac{3c_2\eta\|\mu\|^2}{2} \cdot \frac{|\mathcal{P}|}{n} \left( \frac{|\mathcal{N}|}{|\mathcal{P}|} + \frac{\sqrt{\log(n/\delta)}}{\|\mu\|} \right) \sum_{i \in [n]} \ell_i^{(t)} \\
&\leq 3c_2\eta\|\mu\|^2 \max\left\{ \frac{|\mathcal{N}|}{|\mathcal{P}|}, \frac{\sqrt{\log(n/\delta)}}{\|\mu\|} \right\} \sum_{i \in [n]} \ell_i^{(t)},
\end{aligned}
$$

where $(i)$ follows by Lemma B.2 and $(ii)$ follows by the loss ratio bound in Lemma C.3. Since by assumption $\|\mu\|^2 \geq Cn^2 \log(n/\delta)$, we infer that

$$
\frac{\sqrt{\log(n/\delta)}}{\|\mu\|} \leq \frac{1}{n} \leq \frac{|\mathcal{N}|}{|\mathcal{P}|} = \frac{1}{\tau}.
$$

Thus,

$$
(\theta^{(t+1)} - \theta^{(t)}) \cdot (-\mu_2) \leq 3c_2\eta\|\mu\|^2 \max\left\{ \frac{|\mathcal{N}|}{|\mathcal{P}|}, \frac{\sqrt{\log(n/\delta)}}{\|\mu\|} \right\} \sum_{i \in [n]} \ell_i^{(t)} = \frac{c\eta\|\mu\|^2}{\tau} \sum_{i \in [n]} \ell_i^{(t)},
$$

wrapping up our proof. ∎

With these lemmas in place we are now ready to prove our result. Let us restate it here.

**Theorem 4.3.** *Let $q \sim \mathsf{N}(0, I_{d \times d})$. There exist constants $c$ and $c'$ such that, for all large enough $C$ and for any $0 < \delta < 1/C$, under the assumptions of this subsection the following holds. With probability at least $1 - \delta$, training on $\mathcal{S}$ produces a maximum margin classifier $\widehat{\theta}_{\mathsf{MM}}$ satisfying:*

$$
\mathsf{TestError}[\widehat{\theta}_{\mathsf{MM}}] = \mathbb{P}_{(x,y) \sim \mathsf{P}_{\text{test}}} \left[ \mathrm{sign}\left( \widehat{\theta}_{\mathsf{MM}} \cdot x \right) \neq y \right] \geq \frac{1}{2} \cdot \Phi\left( -\frac{c\sqrt{n}}{\tau} \cdot \frac{\|\mu\|^2}{\sqrt{d}} \right),
$$

*where $\Phi$ is the Gaussian cdf. Furthermore, if the imbalance ratio $\tau \geq c' \frac{\sqrt{n}\|\mu\|^2}{\sqrt{d}}$ then with probability at least $1 - \delta$, $\mathsf{TestError}[\widehat{\theta}_{\mathsf{MM}}] \geq \frac{1}{8}$.*

**Proof** The test error for $\widehat{\theta}_{\mathsf{MM}}$ is

$$
\begin{aligned}
&\mathsf{TestError}[\widehat{\theta}_{\mathsf{MM}}] \\
&= \mathbb{P}_{(x,y) \sim \mathsf{P}_{\text{test}}} \left[ \mathrm{sign}\left( \widehat{\theta}_{\mathsf{MM}} \cdot x \right) \neq y \right] \\
&= \frac{1}{2} \mathbb{P}_{q \sim \mathsf{N}(0, \frac{1}{2} I_{p \times p})} \left[ \mathrm{sign}\left( \widehat{\theta}_{\mathsf{MM}} \cdot (\mu_1 + q) \right) \neq 1 \right] + \frac{1}{2} \mathbb{P}_{q \sim \mathsf{N}(0, \frac{1}{2} I_{p \times p})} \left[ \mathrm{sign}\left( \widehat{\theta}_{\mathsf{MM}} \cdot (\mu_2 + q) \right) \neq -1 \right] \\
&\geq \frac{1}{2} \mathbb{P}_{q \sim \mathsf{N}(0, \frac{1}{2} I_{p \times p})} \left[ \mathrm{sign}\left( \widehat{\theta}_{\mathsf{MM}} \cdot (\mu_2 + q) \right) \neq -1 \right] \\
&= \frac{1}{2} \mathbb{P}_{q \sim \mathsf{N}(0, \frac{1}{2} I_{p \times p})} \left[ -\widehat{\theta}_{\mathsf{MM}} \cdot \mu_2 - \widehat{\theta}_{\mathsf{MM}} \cdot q < 0 \right] \\
&= \frac{1}{2} \mathbb{P}_{q \sim \mathsf{N}(0, \frac{1}{2} I_{p \times p})} \left[ -\widehat{\theta}_{\mathsf{MM}} \cdot q < -\widehat{\theta}_{\mathsf{MM}} \cdot (-\mu_2) \right] \\
&= \frac{1}{2} \mathbb{P}_{\xi \sim \mathsf{N}(0,1)} \left[ \xi < -\widehat{\theta}_{\mathsf{MM}} \cdot (-\mu_2) \right] \quad \text{(since } -\widehat{\theta}_{\mathsf{MM}} \cdot q \sim \mathsf{N}(0,1)) \\
&= \frac{\Phi(-\widehat{\theta}_{\mathsf{MM}} \cdot (-\mu_2))}{2},
\end{aligned}
\tag{23}
$$

where $\Phi$ is the Gaussian cumulative distribution function.

With this inequality in place, we want to prove an upper bound on $\left(\widehat{\theta}_{\mathrm{MM}} \cdot (-\mu_2)\right)^2$. Now, since $\frac{\theta^{(t)}}{\|\theta^{(t)}\|} \to \widehat{\theta}_{\mathrm{MM}}$, it suffices to prove an upper bound on $\left(\lim_{t\to\infty} \frac{\theta^{(t)} \cdot (-\mu_2)}{\|\theta^{(t)}\|}\right)^2$ instead.

To do this, going forward let us assume that a good run (see Definition B.3) occurs. Lemma B.2 guarantees that this happens with probability at least $1 - \delta$.

Let $t_0$ be the constant from Lemma C.2, then by Lemma C.4 we have that for any $t > t_0$

$$\theta^{(t)} \cdot (-\mu_2) = \theta^{(0)} \cdot (-\mu_2) + \sum_{s=1}^{t_0} (\theta^{(s)} - \theta^{(s-1)}) \cdot (-\mu_2) + \sum_{s=t_0+1}^{t} \left(\theta^{(s)} - \theta^{(s-1)}\right) \cdot (-\mu_2)$$

$$\leq \psi + \frac{c_1 \eta \|\mu\|^2}{\tau} \sum_{s=t_0}^{t-1} \sum_{i \in [n]} \ell_i^{(s)}, \tag{24}$$

where we define $\psi := \theta^{(0)} \cdot (-\mu_2) + \sum_{s=1}^{t_0} \left(\theta^{(s)} - \theta^{(s-1)}\right) \cdot (-\mu_2)$.

On the other hand, by repeatedly applying Lemma C.2 we get that

$$\|\theta^{(t)}\| \geq \|\theta^{(t_0)}\| + c_2 \eta \sqrt{\frac{d}{n}} \sum_{s=t_0}^{t-1} \sum_{i \in [n]} \ell_i^{(s)}. \tag{25}$$

Furthermore, since $\|\theta^{(t)}\| \to \infty$ (by Ji et al., 2020, Lemma 2) and

$$\|\theta^{(t+1)}\| = \left\|\theta^{(0)} + \eta \sum_{s=0}^{t} \sum_{i \in [n]} \ell_i^{(s)} z_i\right\| \leq \eta \sum_{s=0}^{t} \sum_{i \in [n]} \ell_i^{(s)} \|z_i\|$$

$$\leq c_3 \eta d \sum_{s=0}^{t} \sum_{i \in [n]} \ell_i^{(s)} \qquad \text{(by Part 1 of Lemma B.2)}$$

$$= c_3 \eta d \sum_{s=0}^{t_0-1} \sum_{i \in [n]} \ell_i^{(s)} + c_3 \eta d \sum_{s=t_0}^{t} \sum_{i \in [n]} \ell_i^{(s)}$$

we can conclude that $\sum_{s=t_0}^{t} \sum_{i \in [n]} \ell_i^{(s)} \to \infty$.

Thus combining inequalities (24) and (25) we get that

$$\lim_{t\to\infty} \frac{\theta^{(t)} \cdot (-\mu_2)}{\|\theta^{(t)}\|} \leq \lim_{t\to\infty} \frac{\psi + \frac{c_1 \eta \|\mu\|^2}{\tau} \sum_{s=t_0}^{t-1} \sum_{i \in [n]} \ell_i^{(s)}}{\|\theta^{(t_0)}\| + c_2 \eta \sqrt{\frac{d}{n}} \sum_{s=t_0}^{t-1} \sum_{i \in [n]} \ell_i^{(s)}} = \frac{c_4}{\tau} \sqrt{\frac{n}{d}} \|\mu\|^2.$$

Plugging this upper bound into inequality (23) completes the proof. ∎

## D  EARLY-STOPPING BEFORE MODELS INTERPOLATE

Prior works (Sagawa et al., 2019; Byrd & Lipton, 2019) found that adding regularization such as strong L2 regularization and early stopping can partially restore the effects of importance weights at the cost of not interpolating the training set. As such, we compare the performance of our polynomially-tailed loss against cross-entropy when models are early-stopped, using the same settings as before. For each run here, we select the model checkpoint with the best weighted validation accuracy and evaluate the checkpoint on the test set. Figure 5 compares the test accuracies of the early-stopped models (IW-ES $(\overline{w})$, IW-ES $(\overline{w}^c)$) against reweighted models trained past interpolation (IW $(\overline{w})$).

Consistent with prior works, we see that early stopping as a form of regularization improves the test accuracies of both loss functions when used with importance weights. Our polynomially-tailed loss

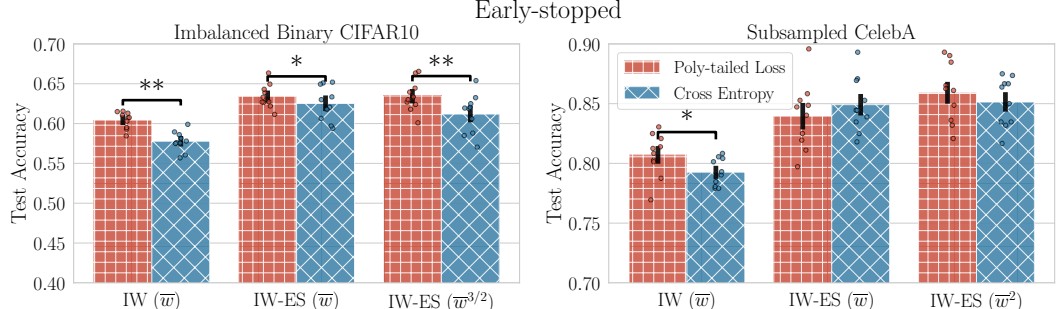

Figure 5: Polynomially-tailed loss versus cross-entropy loss on a label shift dataset and a subpopulation shift dataset for neural networks early-stopped according to the best weighted validation accuracy. $*$ and $**$ indicate $p < 0.05$ and $p < 0.005$ statistical significance, respectively. While early stopping is effective for both losses, our polynomially-tailed loss performs even better on imbalanced binary CIFAR10.

gives test accuracies that are better than or similar to cross-entropy in all weighted loss scenarios. The gain over cross-entropy is statistically significant in the binary CIFAR10 runs. On subsampled CelebA, the polynomially-tailed loss with squared weights attains the highest mean test accuracy out of all settings.

# E  EXPERIMENTAL DETAILS

## E.1  HYPERPARAMETERS

### E.1.1  FOR SECTION 5.1

**Imbalanced binary CIFAR10 experiments.**  We use the same convolutional neural network architecture with random initialization as (Byrd & Lipton, 2019). We train for 400 epochs with SGD with a batch size of 64. We chose hyperparameters that resulted in stable training for each setting. We use a constant 0.001 learning rate with 0.9 momentum for "No IW", "IW ($\overline{w}$)", and "IW-ES ($\overline{w}$)". We use a constant 0.008 learning rate with no momentum for "IW ($\overline{w}^{3/2}$)", and "IW-ES ($\overline{w}^{3/2}$)".

**Subsampled CelebA experiments.**  We use a ResNet50 architecture with ImageNet initialization as done in (Sagawa et al., 2019). Due to the large dataset size, we use only 2% of the full training dataset as mentioned in the main text. Reducing the computation requirements allows us to perform statistical evaluations of the results over sufficiently many seeds. We train for 100 epochs with SGD with a batch size of 64. We chose hyperparameters that resulted in stable training for each setting. We use a constant 0.0004 learning rate with 0.9 momentum for all settings.

### E.1.2  FOR SECTION 5.2

To fairly evaluate every method, we grid search the hyperparamters for each method extensively. We kept the same architectural setup as before, but adjust the optimization settings such that the models train until they interpolate the training data.

**Imbalanced binary CIFAR10 experiments.**

- Cross entropy + Class Undersampling: We use the same hyperparameters as Cross entropy on the full dataset and average results over random undersamplings plus initializations

- LDAM: Following the original paper Cao et al. (2019), we grid search the margin hyperparameter over $[0.1, 0.2, 0.3, 0.4, 0.5, 0.6, 0.7, 0.8, 0.9, 1.0]$. We found $1.0$ to be the best.

- CDT Loss, LA Loss, VS Loss: Following Kini et al. (2021), we tune $\tau$ and $\gamma$ with $\tau = 0$ being CDT loss and $\gamma = 0$ being LA Loss. We grid search across $(\tau, \gamma) \in [0.0, 0.5, 1.0, 1.5, 2.0, 2.5, 3.0] \times [0.0, 0.1, 0.2, 0.3, 0.4, 0.5]$. We found $\tau = 3$ to be the best for LA loss, $\gamma = 0.5$ to be the best for CDT loss, and $\tau = 3, \gamma = 0.3$ to be the best

for VS Loss. Unlike the original paper (Kini et al., 2021), we found that the best hyperparameters of VS loss are similar to those of LA Loss. We initially followed the recommendations of Kini et al. (2021) and grid searched over $(\tau, \gamma) \in [0.5, 0.75, 1.0, 1.25, 1.5] \times [0.05, 0.1, 0.15, 0.2]$, but it gave VS loss poor performance (mean test acc was only 58.6% using the best hyperparameters $\tau = 1, \gamma = 0.05$). Hence, we did a more extensive tuning afterwards.

- Poly-tailed Loss: We grid search over $(\alpha, c) \in [0.25, 0.5, 1.0, 2.0, 4.0] \times [1.0, 1.5, 2.0, 2.5, 3.0]$. We found $\alpha = 2$ and exponent $c = 3$ to be the best.

**Subsampled CelebA.**

- Cross entropy + Group Undersampling: We use the same hyperparameters as Cross entropy on the full dataset and average results over random undersamplings plus initializations.

- VS Loss: We follow the tuning procedure in Kini et al. (2021) for group-sensitive VS Loss and grid search over $\gamma \in [0.1, 0.2, 0.3, 0.4, 0.5, 0.6, 0.7, 0.8, 0.9]$. We found $\gamma = 0.4$ to be the best.

- Poly Loss: We grid search over $(\alpha, c) \in [1, 2, 3] \times [1.0, 1.5, 2.0, 2.5, 3.0]$. We found $\alpha = 2$ and exponent $c = 2.5$ to be the best.

- Cross entropy + DRO: We tune the adversarial step size $\eta_q$ over $[0.0025, 0.005, 0.01, 0.02, 0.03, 0.04, 0.05, 0.1]$. We found $\eta_q = 0.05$ to be the best.

- VS Loss + DRO: We used the best $\gamma$ from VS Loss alone and $\eta_q = 0.05$ for DRO.

- Poly Loss + DRO: We used the best $\alpha$ from Poly Loss alone and $\eta_q = 0.05$ for DRO.

### E.2 IMPORTANCE WEIGHT COMPUTATION.

Here we explain in greater detail how we compute the importance weights. Note that the scale of the importance weights is important in ensuring stable training. Let there be $G$ groups of subpopulations in the training and testing set. $G = 2$ for binary CIFAR10 and $G = 4$ for CelebA. Let $n_1, \ldots, n_G$ be the counts of each group in the training set. Let $n$ be the total number of training examples. In our case, the test set is balanced across the groups. Our procedure for computing weights is:

1. For each example $i$ that belongs to group $g$, compute the unbiased weight: $\overline{w}_i \leftarrow 1/n_g$.

2. Exponentiate the weights by $c$ if necessary: $w_i \leftarrow \overline{w}_i^c$. No importance weighting corresponds to $c = 0$. Unbiased importance weighting corresponds to $c = 1$.

3. To adjust for the scale of weights, normalize by the average exponentiated weight across the *full* training set: $w_i \leftarrow w_i / \sum_{j=1}^{n} w_i$. Only normalizing across each minibatch can result in unstable training, especially when a minibatch contains no representatives from a group.

### E.3 EARLY-STOPPING METRIC.

We use the checkpoint with the highest importance weighted validation accuracy when early stopping. We compute separate importance weights of the validation set with respect to the test set, and reweight the validation accuracy using *unbiased* weights, even when the training weights are biased. Our procedure allows for the situation where the validation set is not exactly the same distribution as the training set.

E.4   EXACT NUMERICAL RESULTS OF OUR EXPERIMENTS

| Dataset | | Setting | Cross Entropy mean | std. err. | Poly-tailed Loss mean | std. err. | $p$-value |
|---|---|---|---|---|---|---|---|
| Subsampled CelebA | Early-stopped | IW | 79.3% | 0.4% | 80.7% | 0.6% | 0.014 |
| | | IW-ES | 84.9% | 0.7% | 84.0% | 0.9% | 0.762 |
| | | IW-Exp-ES | 85.1% | 0.7% | 85.9% | 0.8% | 0.202 |
| | Interpolating | IW | 79.3% | 0.4% | 80.7% | 0.6% | 0.014 |
| | | IW-Exp | 78.7% | 0.4% | 82.7% | 0.4% | 0.000 |
| | | No IW | 79.6% | 0.4% | 78.5% | 0.4% | 0.999 |
| Binary CIFAR10 | Early-stopped | IW | 57.8% | 0.4% | 60.4% | 0.3% | 0.000 |
| | | IW-ES | 62.5% | 0.7% | 63.4% | 0.5% | 0.022 |
| | | IW-Exp-ES | 61.2% | 0.8% | 63.5% | 0.6% | 0.001 |
| | Interpolating | IW | 57.8% | 0.4% | 60.4% | 0.3% | 0.000 |
| | | IW-Exp | 57.4% | 0.6% | 63.0% | 0.6% | 0.000 |
| | | No IW | 60.5% | 0.8% | 56.4% | 0.4% | 1.000 |

Table 1: Numerical results corresponding to Figure 3 and 5. Here "Exp" corresponds to exponentiated weights. We use $3/2$ and $2$ as the exponents in Binary CIFAR10 and CelebA respectively. We use $\alpha = 1$ for all of these experiments without tuning $\alpha$.

| Setting | Summary of method | Best settings | Mean test acc | Std err |
|---|---|---|---|---|
| Cross Entropy | | N/A | 60.5% | 0.8% |
| Cross Entropy + IW | Classical importance weighting | N/A | 57.8% | 0.4% |
| Cross Entropy + Class undersampling | Balance the classes by throwing away majority data | N/A | 58.5% | 1.4% |
| LDAM | Add class dependent constants to logits | largest margin=1.0 | 58.7% | 1.6% |
| CDT | Multiply class dependent constants by logits | $\gamma = 0.5$ | 60.2% | 1.6% |
| LA Loss | Add class dependent constants to logits | $\tau = 3$ | 66.9% | 2.1% |
| VS Loss (best setting around range suggested by paper) | Multiply by and add class dependent constants to logits | $\tau = 1, \gamma = 0.05$ | 58.6% | 1.4% |
| VS Loss (our own tuning) | Multiply by and add class dependent constants to logits | $\tau = 3, \gamma = 0.3$ | **69.3%** | 0.9% |
| Poly-tailed Loss | Change loss function and exponentiate importance weights | $\alpha = 2, \overline{w}^3$ | 67.3% | 0.8% |

Table 2: Results on imbalanced binary CIFAR10, corresponding to Figure 4 (left).

| Setting | Method summary | Best settings | Mean test acc | Std err | Worst group test acc | Std err |
|---|---|---|---|---|---|---|
| Cross Entropy | | N/A | 79.6% | 1.2% | 43.1% | 2.9% |
| Cross Entropy + Group undersampling | Balance the groups by throwing away overrepresented data | N/A | 82.3% | 3.2% | **72.1%** | 8.6% |
| Cross Entropy + IW | Classical importance weighting | N/A | 79.3% | 0.4% | 43.1% | 2.9% |
| VS Loss | Multiply by and add class dependent constants to logits | $\gamma = 0.4$ | **82.6%** | 0.7% | 51.3% | 1.5% |
| Poly-tailed Loss | Change loss function and exponentiate importance weights | $\alpha = 2$, $\overline{w}^{2.5}$ | 82.4% | 0.8% | 53.3% | 1.5% |
| Cross Entropy + DRO | | N/A | 84.8% | 0.9% | 60.4% | 2.4% |
| VS Loss + DRO | Change loss function and use DRO | $\gamma = 0.4$ | 83.2% | 0.9% | 58.2% | 2.0% |
| Poly-tailed Loss + DRO | Change loss function and use DRO | $\alpha = 2$ | **86.7%** | 1.1% | **66.5%** | 1.9% |

Table 3: Results on subsampled CelebA, corresponding to Figure 4 (right)

