# OpenReview forum: "Is Importance Weighting Incompatible with Interpolating Classifiers?"
_ICLR.cc/2022/Conference — ICLR 2022 Poster_

### Official Review · Reviewer_s5yg · 2021-11-02

**Correctness:** 3
**Technical Novelty And Significance:** 3
**Empirical Novelty And Significance:** 3
**Recommendation:** 8
**Confidence:** 3

**Main Review:**

[Strengths]
- Clear presentation. Figure 1 gives an intuitive example illustrating why importance weighting could fail on exponentially-tailed losses.
- The proposed polynomially-tailed loss could be a very practical tool for importance weighting.
- Theoretical results look solid and well justify the polynomially-tailed loss.

[Weaknesses]
- The theoretical results are based on the assumption of sub-Gaussian data which could be unrealistic.
- The polynomiallhy-tailed loss does not seem to admit a probabilistic interpretation as the distance between two distributions. How would it perform on a softmax layer?
- The proposed loss has a parameter alpha. However, the impact of the parameter is not discussed in either the theoretical or empirical section. Does the choice of alpha significantly affect the result?



**Summary Of The Paper:**

In this paper, the author investigates whether importance weighting is incompatible with the training of overparameterized neural networks. In contrast to the recent observation that  importance weighting is ineffective in current deep learning paradigm (Byrd & Lipton 2019), the authors show that it could actually be helpful using polynomially-tailed losses. Both theoretical justifications and empirical evidences are provided to support the claim.

**Summary Of The Review:**

Overall, I think the paper provides answers to an important question. Both theoretical and empirical results are provided to corroborate the claim.

---

> ### Author Response · Authors · 2021-11-17
> **Addressing your concerns**
>
> > The theoretical results are based on the assumption of sub-Gaussian data which could be unrealistic.
>
> We agree that extending our results to richer data distributions would indeed be very interesting. However, even in the absence of any distribution shift the generalization analysis of overparameterized linear classifiers in this sub-Gaussian cluster setting is an active area of research recently: (Chatterji & Long, 2021; Wang & Thrampoulidis, 2021; Liang & Recht, 2021; Cao et al., 2021). We therefore study linear models to perform a fine-grained analysis to garner intuition that transfers over to non-linear models.
>
> > The polynomially-tailed loss does not seem to admit a probabilistic interpretation…
>
> We chose the poly-tailed losses since they admit the right implicit bias when trained with gradient descent. However, one can also easily derive the log-likelihood corresponding to the loss by using the relation in Buet-Golfouse (2021), Section 4.1 (http://proceedings.mlr.press/v139/buet-golfouse21a/buet-golfouse21a.pdf). One can also see that the examples in Section 5.3 of this paper shows that there are other polynomially tailed losses fulfilling our condition that follow from taking a latent variable interpretation with Cauchy or Student-t distributions.
>
> > Does the choice of alpha significantly affect the result?
>
> Though we did not tune $\alpha$ in our original experiments, we do so in our new experiments. Tuning alpha can indeed improve performance significantly. Please see our top level response on choosing $\alpha$.
> We analyze the theory for $\alpha=1$, similar analyses can be done for other values of $\alpha$.

---

### Official Review · Reviewer_DWSu · 2021-11-04

**Correctness:** 4
**Technical Novelty And Significance:** 3
**Empirical Novelty And Significance:** 3
**Recommendation:** 6
**Confidence:** 4

**Main Review:**

#### Strengths

1. This paper points out a potential way to make training overparameterized neural nets compatible with importance sampling, which is to change the loss from cross-entropy to a polynomially-tailed loss.
2. A full characterization for the implicit bias of GD on polynomially-tailed loss is given by (2) in Section 4.1 (although it is an easy corollary of Ji et al. 2020). It is clear from (2) that the implicit bias changes with the importance weights.
3. The separation between polynomially-tailed loss and exponentially-tailed loss in Section 4.2 is simple and interesting.

#### Weaknesses

1. The theoretical analysis is only for linear classification. As for the generalization aspect, the analysis is even more restricted as it only holds for the dataset with two sub-Gaussian clusters. So the theoretical result is not very strong. Although we can certainly draw some insights from this simple setting, it is still unclear what generalization guarantees polynomially-tailed loss has in general.
2. The paper claims that the implicit bias of GD changes with importance weights, but such change could be minor when the degree of the polynomial is high, i.e., $\alpha$ is large. In the equation (2) in Section 4.2, if $\alpha \to \infty$, then $\hat{\theta}_{\alpha}$ is just the max-margin solution. Thus the choice of $\alpha$ cannot be arbitrary, and a polynomially-tailed loss with a high degree can perform as badly as the exponential loss. In the later sections, the authors always use $\alpha = 1$. It would be nice if the authors can carefully discuss more the choice of $\alpha$ and provide a recommendation for the choice of $\alpha$. If $\alpha = 1$ is the best in the theoretical setting in Section 4.2 and in the experiments, then the paper should highlight linearly-tailed loss rather than polynomially-tailed loss.
3. The experiment results are not compared against many baselines. It is unclear whether the 2%-3% test accuracy difference can be considered to be a big gain, and whether other methods (e.g., undersampling) can obtain the same gain in terms of the test accuracy. Although this paper is mainly a theoretical paper, the authors do claim that polynomially-tailed loss has practical value, and the current experiment results are not strong enough to support this claim.

#### Minor Comments

* Page 20 looks wried.
* The current proof for Theorem 4.3 is trajectory-based, i.e., it analyzes the trajectory. However, Soudry et al. (2018) have already proved that the convergent classifier is the max-margin classifier, and one can easily show that the max-margin classifier is unique. So I'm wondering whether Theorem 4.3 can be proved by just analyzing the properties of the max-margin classifier, without analyzing the trajectory. It is not a weakness of this paper, but if this paper can provide an argument directly for the max-margin classifier, then the readers will better understand why the max-margin classifier is bad in this setting. After skimming the proof, I think it might be possible to translate Lemma C.3 into a lemma on the upper bounds for the ratio between dual variables (instead of loss values), and it sounds promising to use this upper bound to bound the influence of the samples from the minority class, and hence the max-margin classifier generalizes poorly.
* Section 4.1 should clarify that the optimal solution for (2) is unique. Although it is proved in the appendix that the solution is unique, but clarifying this in the main body can avoid confusion.

**Summary Of The Paper:**

Previous works pointed out an interesting incompatibility between importance sampling and training overparameterized neural nets. In the view of margin, Soudry et al. (2018) showed that GD on exponentially-tailed loss converges to the max-margin classifier, and Xu et al. (2020) pointed out that this margin maximization is unaffected by importance sampling. This paper then turns to studying polynomially-tailed loss.

* This paper establishes the implicit bias by formulating an optimization problem, which can change according to the importance weights. The convergent classifier is shown to generalize better than the exponentially-tailed loss in a simple linear classification setting, where the dataset contains two sub-Gaussian clusters.
* Experiments on Imbalanced Binary CIFAR10 and CelebA show that the polynomially-tailed loss performs better on imbalanced datasets.

**Summary Of The Review:**

This paper provides interesting theoretical results for using polynomially-tailed loss to resolve the incompatibility between importance sampling and training overparameterized neural nets. However, the theoretical setting is quite restrictive. I think this paper is on the borderline, but it is ok to accept this paper.

------------------
Post-rebuttal update:

I am glad to see that the authors have conducted more experiments to compare their method with baselines. I'm not an expert in the empirical methods for dealing with distribution shift, but I think the current empirical results they have are great additions to their theoretical analysis. I have increased my score for the empirical novelty and significance accordingly.
As for the theoretical analysis, which is the main selling point of this paper, I still believe the current setting is restrictive, although I know some previous works are studying it. Therefore, I would like to continue to vote for weak acceptance.

---

> ### Author Response · Authors · 2021-11-17
> **Addressing your concerns**
>
> > The theoretical analysis is only for linear classification. As for the generalization aspect, the analysis is even more restricted as it only holds for the dataset with two sub-Gaussian clusters.
>
> We agree that extending these results to non-linear models and richer data distributions would indeed be very interesting. However, even in the absence of any distribution shift the generalization analysis of overparameterized linear classifiers is an active area of research recently: (Chatterji & Long, 2021; Wang & Thrampoulidis, 2021; Liang & Recht, 2021; Cao et al., 2021). We therefore study linear models to perform a fine-grained analysis to garner intuition that transfers over to non-linear models.
>
> > It would be nice if the authors can carefully discuss more the choice of alpha
>
> We agree, please see our top level response on choosing $\alpha$.
>
> > The experiment results are not compared against many baselines
>
> We have now trained over 450 models to compare to many strong baselines, including undersampling, as you suggested. Our method performs as well or better than prior distribution shift methods, while undersampling remains a very strong baseline on CelebA (results are presented in our top level OpenReview response)
>
> > I'm wondering whether Theorem 4.3 can be proved by just analyzing the properties of the max-margin classifier, without analyzing the trajectory
>
> This is an interesting idea, and perhaps it is indeed possible to study the test error of the maximum-margin classifier without analyzing the trajectory. Here, since we do not have a closed-form expression for the maximum-margin classifier, directly analyzing the iterates of gradient descent allowed us to derive tight lower bounds on the test error.

---

### Official Review · Reviewer_AUFR · 2021-11-08

**Correctness:** 4
**Technical Novelty And Significance:** 2
**Empirical Novelty And Significance:** 2
**Recommendation:** 5
**Confidence:** 4

**Main Review:**

Strengths:
- The paper is well-written and enjoyable to read
- The theoretical claims are sound and well presented
- Theorem 4.2 is interesting: It extends the analysis by Chatterji & Long to imbalanced setting and the finding about the scaling of the weight being cubic in \tau is interesting.

Weaknesses:
- The study is limited to binary classification. It is not clear how to extend the loss to multiclass. At least when it comes to experimental results, multiclass experiments would make a stronger case.
- Experimental results are limited to a comparison between weighted cross entropy and the new loss. How does the new loss compare to other const-sensitive techniques, e.g. LDAM loss [1], Logit-adjusted loss [2], VS loss [3], DRO (Sagawa et al.)?
- More generally, while the presentation is from the viewpoint of distribution shift, all study cases (both theoretical and experimental) are with respect to imbalanced data problems. In this setting there has been a lot of recent work, which appears closely related but is not discussed (eg. [1-4]). Specifically, [3] appears to be very closely related (for example, see their Theorem 1; how does your program in (2) compare to the cost-sensitive svm?). I suggest that the authors try to better place their work within this literature.

[1] Cao, Kaidi, et al. "Learning imbalanced datasets with label-distribution-aware margin loss." arXiv preprint arXiv:1906.07413 (2019).
[2] Menon, Aditya Krishna, et al. "Long-tail learning via logit adjustment." arXiv preprint arXiv:2007.07314 (2020).
[3] Kini, Ganesh Ramachandra, et al. "Label-Imbalanced and Group-Sensitive Classification under Overparameterization." arXiv preprint arXiv:2103.01550 (2021).
[4] Kang, Bingyi, et al. "Decoupling representation and classifier for long-tailed recognition." arXiv preprint arXiv:1910.09217 (2019).

**Summary Of The Paper:**

The paper shows both theoretically and experimentally that importance weighting is not incompatible  with the training of overparameterized models provided that the training loss is appropriately modified so that it does not have an exponential tail. Specifically, for binary classification the authors propose a new loss function with polynomial tail decay. Theoretically, the new loss is shown to outperform weighted cross entropy for linear models and an imbalanced mixtures of gaussian model. Empirically, the new loss outperforms weighted cross entropy on imbalanced CIFAR 10 and the CelebA dataset.

**Summary Of The Review:**

While I appreciate the author's clarity of presentingtheir ideas/theories/proofs/experiments and the paper's finding is quite interesting, it is perhaps not so surprising especially given the recent works and findings on mitigating the effect of weighted CE failure for imbalanced data. Also, the implications of the finding are not made clear. How does the proposed loss compare to state-of-the-art loss adjustments? Does it apply to multiclass settings?

---

> ### Author Response · Authors · 2021-11-17
> **Addressing your conerns**
>
> > Experimental results are limited to a comparison between weighted cross entropy and the new loss. How does the new loss compare to other const-sensitive techniques, e.g. LDAM loss [1], Logit-adjusted loss [2], VS loss [3], DRO (Sagawa et al.)?
>
> Thank you for pointing us to [3], which we missed due to its recency. We will be sure to include comparisons and citations to  [1-3] in the revised draft. We did not compare to the other references originally since we were focused on showing our method fixes importance weighting for interpolating models. We have trained over 450 models to compare to all your suggested baselines, showing that our method performs comparably to or sometimes even exceeding recent distribution shift methods (results are presented in our top level OpenReview response).
>
> > … while the presentation is from the viewpoint of distribution shift, all study cases (both theoretical and experimental) are with respect to imbalanced data problems
>
> - In our experiments we consider two types of distribution shift: label shift (CIFAR10) and sub-population shift dataset (CelebA).
> - We believe that our theoretical analysis of generalization (Theorem 4.2 and 4.3) can potentially be extended to analyze the spurious correlations setting studied by Sagawa et al. (2020) (http://proceedings.mlr.press/v119/sagawa20a.html).
>
>
>
> > The study is limited to binary classification. It is not clear how to extend the loss to multiclass
>
> Extensions to multiclass classification would indeed be interesting. We note that a polynomially tailed loss can be obtained by suitably modifying the cross-entropy loss.
>
>
> > [the paper's finding] is perhaps not so surprising especially given the recent works and findings on mitigating the effect of weighted CE failure for imbalanced data
>
> Our motivation aims to directly revive importance weighting as an effective technique for training interpolating classifiers under distribution shift. Previous techniques that multiply terms to the logits do not constitute importance weighting methods, whereas techniques that add terms to the logits do not change the asymptotic behavior of gradient descent. Our proposed loss is a fundamentally new loss function with new surprising properties, e.g. the optimal weights is the imbalance ratio raised to a power.

---

> > ### Comment · Reviewer_AUFR · 2021-11-30
> > **Thank you for the responses**
> >
> > I thank the authors for their responses. I appreciate their effort and time on the additional experimental results. The additional discussion and comparisons to [1-5] are indeed strengthening the paper. I am increasing my score (with my main concern making it borderline the lack of a multiclass version). Overall, I am ok with the paper being accepted once the proposed edits are made.

---

### Official Review · Reviewer_46YL · 2021-11-11

**Correctness:** 4
**Technical Novelty And Significance:** 3
**Empirical Novelty And Significance:** 3
**Recommendation:** 6
**Confidence:** 4

**Main Review:**

This paper studies the impact of importance-weighting in interpolating classifiers. The authors showed that in contrast to the prior works that showed importance-weighting is ineffective due to implicit bias of converging to max-margin, switching to poly-tailed loss circumvents this issue, both theoretically and empirically.

Pros:
Overall, this paper is very well-written and the theoretical results are sound. I really like the idea of using a poly-tailed loss - it is simple, effective, and practical - but by far not an obvious one. This idea circumvents the previous negative results in an elegant manner.

I also like the analysis of poly-tailed loss under Gaussian mixture setting: there are some surprising takeaways, e..g. inverse-frequency weighting being sub-optimal and the optimal re-weighting being roughly cubic in class imbalance. I have never seen these types of results in the prior works.

Cons:

There are some obvious limitations, for example, the gap between linear classifiers and neural networks used in practice, which is a minor issue due to the intractability of analysis.

There are some more interesting possible extensions, though: for example, a theoretical analysis of (and comparing against) early stopping would be nice.

There is limited discussion about how to pick the hyperparameter alpha, which can be useful for practitioners.

The performance boost (2%-3%) is relatively minor. I am not sure about how it compares to stronger baselines.

**Summary Of The Paper:**

This paper studies the impact of importance-weighting in interpolating classifiers. The authors showed that in contrast to the prior works that showed importance-weighting is ineffective due to implicit bias of converging to max-margin, switching to poly-tailed loss circumvents this issue, both theoretically and empirically.

**Summary Of The Review:**

I like the paper due to its simplicity, effectiveness, and interesting take-aways from theory. There is some room for improvement in the experiments.

---

> ### Author Response · Authors · 2021-11-17
> **Addressing your concerns**
>
> > There is limited discussion about how to pick the hyperparameter alpha, which can be useful for practitioners.
>
> We agree, please see our top level response on choosing $\alpha$.
>
> > The performance boost (2%-3%) is relatively minor. I am not sure about how it compares to stronger baselines.
>
> Good point! We have trained over 450 models to compare to a variety of strong baselines, showing that our method performs as well or better than prior distribution shift methods (results are presented in our top level OpenReview response). With proper tuning, our loss function can increase performance by 10% or more compared to cross entropy.
>
>
> > There are some more interesting possible extensions, though: for example, a theoretical analysis of (and comparing against) early stopping would be nice.
>
> The theoretical comparison between early-stopping and poly loss is indeed very interesting and something we have been thinking about. We analyzed a simplified data model where all centered feature vectors have the same norm and are mutually orthogonal, as opposed to the approximate orthogonality obtained in Lemma B.2. We have preliminary results showing that the test error of optimally early-stopping on cross-entropy loss scales as $\exp(-c\lVert \mu \rVert^{4}/d)$, which is the same as the upper bound for test error on poly loss with cubic weights. Under this data model, we can also further refine the upper bound for the importance weighted poly loss classifier to $\exp(-c(\lVert \mu \rVert^2 n) / (d \cdot \tau))$, which is tighter than the current bound in Theorem 4.2. While this analysis is on a much simplified data model, we believe that it is a reasonable approximation of the overparameterized regime with sub-Gaussian data, so we expect to be able to generalize the results. If these claims hold in general, then it will be shown that cubic weighted poly loss outperforms early-stopped cross-entropy when the imbalance $\tau=o(n)$.

---

> > ### Author Response · Authors · 2021-12-04
> > **We hope our rebuttal addressed your concerns**
> >
> > We wanted to reach out once again and ask whether our rebuttal addressed your concerns. It felt to us that your main criticism of the main paper was the lack of detailed empirical evaluation. We have thoroughly addressed this in our rebuttal (see our top-level comment in OpenReview) and we hope that you can raise your score as a result.

---

### Author Response · Authors · 2021-11-17
**Summary comment**

We thank all reviewers for their feedback, which has helped us improve our paper. We emphasize that our paper seeks to *understand* the failure of importance weighting for interpolating models, as observed by prior work, and provide a remedy for the procedure. Our goal was not to claim our method exceeds state of the art on imbalance data correction. However, at the reviewers' suggestion, we have performed comparisons to existing SoTA methods, and we show that our method (as well as simple baselines such as undersampling, early stopping) generally match the performance of these methods (see tables below).

There are many imbalanced data correction methods that modify the logits (e.g. [1-4]), but they do not address our movitivating scientific question:
>*Can weights on training losses viably correct distribution shifts for interpolating networks?*

We answer this question affirmatively with clear theoretical evidence and competitive empirical results.

---

> ### Author Response · Authors · 2021-11-17
> **Empirical comparisons to other works**
>
> We agree that placing our method in the context of prior distribution shift methods would greatly strengthen our paper. We *extensively* tune and compare to all methods suggested by the reviewers: undersampling as reviewer DWSu suggested, and the cost-sensitive methods that reviewer AUFR listed [1-4], which we will also cite. We also compare each loss function with and without DRO [5]. For each method, we fix the seed and grid search thoroughly on the validation set. We then train the best setting, averaging over 10 model initializations. We set the training epochs such that all methods attained 100% train accuracy. Previously, we did not tune $\alpha$, but instead always set it equal to 1. **After tuning all methods, our new experiments show that poly-tailed loss with exponentiated weights gives accuracies comparable to or even exceeding those by recently proposed methods.**
>
> ## CelebA
>
> | Name                                   | Summary of method                                            | Best settings                | Mean test acc     | Std err   | Worst group test acc  | Std err   |
> |------                                  |-------------------                                           |---------------               |------             |-----      |------                 |-----      |
> | Cross Entropy                          |                                                              | N/A                          | 79.6              | 1.2       | 43.1                  | 2.9       |
> | Cross Entropy + Group undersampling    | Balance the groups by throwing away overrepresented data     | N/A                          | 82.3              | 3.2       | **72.1**              | 8.6       |
> | Cross Entropy + IW                     | Classical importance weighting                               | N/A                          | 79.3              | 0.4       | 43.1                  | 2.9       |
> | VS Loss [3]                            | Multiply by and add class dependent constants to logits      | $\gamma=0.4$                 | 82.6              | 0.7       | 51.3                  | 1.5       |
> | Poly-tailed Loss                       | Change loss function and exponentiate importance weights     | $\alpha=2, \rho=2.5$         | 82.4              | 0.8       | 53.3                  | 1.5       |
> | -----                                  | -----                                                        |-----                         |-----              |-----      |-----                  |-----      |
> | Cross Entropy + DRO [5]                |                                                              | N/A                          | 84.8              | 0.9       | 60.4                  | 2.4       |
> | VS Loss + DRO                          | Change loss function and use DRO                             | $\gamma=0.4$                 | 83.2              | 0.9       | 58.2                  | 2.0       |
> | Poly-tailed Loss + DRO                 | Change loss function and use DRO                             | $\alpha=2$                   | **86.7**          | 1.1       | **66.5**              | 1.9       |
>
> On CelebA, we compare mostly to VS loss since VS loss encapsulates LA loss, CDT loss, and LDAM loss. Poly-tailed loss with IW achieves comparable test accuracy to VSLoss. When we use both loss functions with DRO, poly loss + DRO is better than VSLoss + DRO by 3.5% on test accuracy and 8% better on worst group accuracy.
> For all methods, including ones that use DRO, we train all models to interpolation without early stopping or strong regularizations, since we are interested in interpolating classifiers. We emphasize that *we expect poly loss with IW to perform well on average test accuracy*, and that poly-tailed with IW does not target worst group accuracy directly in the training objective. However, in practice poly-tailed loss with IW gives the best worst group accuracy out of all loss functions.
>
> ### Tuning procedure
>
> - Cross entropy + undersampling by group: we use the same hyperparameters as Cross entropy on the full dataset and average results over random undersamplings plus initializations.
>
> - VS Loss: We follow the tuning procedure in [3] for group-sensitive VS loss and grid search over $\gamma \in \[0.1, 0.2, 0.3, 0.4, 0.5, 0.6, 0.7, 0.8, 0.9\]$. We found $\gamma=0.4$ to be best.
>
> - Poly Loss: We grid search over $(\alpha, \rho) \in \[1, 2, 3\] \times \[1.0, 1.5, 2.0, 2.5, 3.0\]$. We found $\alpha=2, \rho=2.5$ to be best.
> Cross entropy + DRO: We tune the adversarial step size $\eta_q$ over $\[0.0025, 0.005, 0.01, 0.02, 0.03, 0.04, 0.05, 0.1\]$. We found that $\eta_q=0.05$ to be best.
>
> - VS Loss + DRO: We used the best $\gamma$ from VS loss without DRO with $\eta_q=0.05$
>
> - Poly Loss + DRO: We used the best $\alpha$ from Poly Loss without DRO with $\eta_q=0.05$

---

> > ### Author Response · Authors · 2021-11-17
> > **Empirical comparisons to other works (cont'd)**
> >
> > ## Imbalanced Binary CIFAR10
> > | Name                                                       | Summary of method                                        | Best settings          | Mean test acc      | Std err   |
> > |-----                                                       |-------------------                                       |---------------         |------              |------     |
> > | Cross Entropy                                              |                                                          | N/A                    | 60.5               | 0.8       |
> > | Cross Entropy + IW                                         | Classical importance weighting                           | N/A                    | 57.8               | 0.4       |
> > | Cross Entropy + Class undersampling                        | Balance the classes by throwing away majority data       | N/A                    | 58.5               | 1.4       |
> > | LDAM [1]                                                   | Add class dependent constants to logits                  | largest margin=1.0     | 58.7               | 1.6       |
> > | CDT [4]                                                    | Multiply class dependent constants by logits             | $\gamma=0.5$           | 60.2               | 1.6       |
> > | LA Loss [2]                                                | Add class dependent constants to logits                  | $\tau=3$               | 66.9               | 2.1       |
> > | VS Loss (best setting around range suggested by [3])       | Multiply by and add class dependent constants to logits  | $\tau=1, \gamma=0.05$  | **58.6**           | 1.4       |
> > | VS Loss (our own tuning)                                   | Multiply by and add class dependent constants to logits  | $\tau=3, \gamma=0.3$   | **69.3**           | 0.9       |
> > | Poly-tailed Loss                                           | Change loss function and exponentiate importance weights | $\alpha=2,\rho=3$      | 67.3               | 0.8       |
> >
> > On binary CIFAR10, our poly-tailed loss performs comparably to LA Loss and is only 2% worse than VS loss in the mean test accuracy with overlapping confidence intervals. Note that we grid searched across the published hyperparameter ranges from [3] when we first tuned VS loss, and it performed poorly (58.6% test accuracy using the best hyperparameters $\tau=1, \gamma=0.05$). After a more thorough grid search for VS loss, we found optimal hyperparameters that were similar to those of LA loss, increasing test accuracy by 10%.
> > In sum, our poly-tailed loss matched or exceeded existing data imbalance adjustment methods (under published hyperparameters) and was beaten by 2% only after more extensive hyperparameter tuning on our part.
> >
> > ### Tuning procedure
> > - Cross entropy + undersampling by class: we use the same hyperparameters as Cross entropy on the full dataset and average results over random undersamplings plus initializations
> > - LDAM: Following the original paper [1], we grid search the margin hyperparameter over $\[0.1, 0.2, 0.3, 0.4, 0.5, 0.6, 0.7, 0.8, 0.9, 1.0\]$.
> > - CDT Loss, LA Loss, VS Loss:  Following [3], we tune $\tau$ and $\gamma$ with $\tau=0$ being CDT loss and $\gamma = 0$ being LA Loss. We grid search across $(\tau, \gamma) \in \[0.0, 0.5, 1.0, 1.5, 2.0, 2.5, 3.0\] \times \[0.0, 0.1, 0.2, 0.3, 0.4, 0.5\]$. Unlike the original paper [3], we found that the best hyperparameters of VS loss are similar to those of LA Loss. We initially followed the recommendations of [3] and grid searched over $(\tau, \gamma) \in \[0.5, 0.75, 1.0, 1.25, 1.5\] \times \[0.05, 0.1, 0.15, 0.2\]$, but it gave VS loss poor performance (mean test acc = 58.6).
> > - Poly-tailed Loss: We grid search over $(\alpha, \rho) \in \[0.25, 0.5, 1.0, 2.0, 4.0\] \times \[1.0, 1.5, 2.0, 2.5, 3.0\]$.

---

> > > ### Author Response · Authors · 2021-11-17
> > > **Theoretical comparisons to other works**
> > >
> > > While the multiplicative logit-adjustment losses (VS loss and CDT loss [3]) do affect the implicit bias of the final learned classifier, they do not demonstrate that reweighted losses can solve dataset imbalance. Even if existing results address this, the theory in our work provides stronger results in several ways:
> > > - VS loss theory [3] guarantees balanced test error across groups asymptotically in $n$ and $d$ for some optimal hyperparameter.
> > > - This leaves open several important questions about the optimality of distribution shift correction which we address in our work.
> > >     - First, can we achieve learning at the information theoretic limit of $\lVert \mu \rVert^2 / d \to \infty$? Our work shows poly loss attains this (Thm 4.2).
> > >     - Second, even if group errors are balanced, the errors for the groups may still be high. Our work shows poly loss attains low per-group error, not just balanced error and that this exceeds the performance of max-margin predictors (Thms 4.2, 4.3)
> > >     - Finally, the result for VS loss relies on the existence of some, unknown hyperparameters which may vary greatly across different problem instances (as shown in our CIFAR-10 experiments), we show that for poly-loss importance weighting in the linear setting, a single fixed universal hyperparameter can perform well across instances ($w=\tau^3$, Thm 4.2).

---

> > > > ### Author Response · Authors · 2021-11-17
> > > > **Choice for $\alpha$**
> > > >
> > > > - Though we analyze the theory for $\alpha=1$, similar analyses can be done for other values of $\alpha$.
> > > > - We did not empirically tune $\alpha$ in our original experiments, using $\alpha=1$ always. However, in the new results above, we tune over $\alpha$ and the weight exponent to match the fact that LA/CDT/VS baselines tune hyperparameters. Our tuning procedures described above can provide some additional guidelines on reasonable ranges for $\alpha$ empirically. We found training to be less stable for $\alpha < 1$.

---

> > > > > ### Author Response · Authors · 2021-11-17
> > > > > **References mentioned**
> > > > >
> > > > > [1] Cao, Kaidi, et al. "Learning imbalanced datasets with label-distribution-aware margin loss." arXiv preprint arXiv:1906.07413 (2019).
> > > > >
> > > > > [2] Menon, Aditya Krishna, et al. "Long-tail learning via logit adjustment." arXiv preprint arXiv:2007.07314 (2020).
> > > > >
> > > > > [3] Kini, Ganesh Ramachandra, et al. "Label-Imbalanced and Group-Sensitive Classification under Overparameterization." arXiv preprint arXiv:2103.01550 (2021).
> > > > >
> > > > > [4] Kang, Bingyi, et al. "Decoupling representation and classifier for long-tailed recognition." arXiv preprint arXiv:1910.09217 (2019).
> > > > >
> > > > > [5] Sagawa, et al. “Distributionally Robust Neural Networks for Group Shifts: On the Importance of Regularization for Worst-Case Generalization” arXiv preprint arXiv:1911.08731 (2019).

---

### Decision · Program_Chairs · 2022-01-20

**Decision:**

Accept (Poster)

**Comment:**

The paper revisits importance sampling as an approach for combating distribution shift when training over-parameterized neural networks. Contrary to recent results that suggest that importance sampling is perhaps incompatible with over-parameterization, the authors find that the exponential tail of losses such as the logistic loss is the root cause. For polynomial tailed losses, authors analyze gradient descent on importance weighted polynomially-tailed losses and demonstrate the advantage of importance sampling in a label shift setting.

There paper is well-written and the results are sound. Overall, a good paper.